# Molecular mechanism of influenza A NS1-mediated TRIM25 recognition and inhibition

Marios G. Koliopoulos[1], Mathilde Lethier[2], Annemarthe G. van der Veen[3], Kevin Haubrich[4], Janosch Hennig[4], Eva Kowalinski[2], Rebecca V. Stevens[1], Stephen R. Martin[5], Caetano Reis e Sousa [3], Stephen Cusack [2] & Katrin Rittinger [1]

RIG-I is a viral RNA sensor that induces the production of type I interferon (IFN) in response to infection with a variety of viruses. Modification of RIG-I with K63-linked poly-ubiquitin chains, synthesised by TRIM25, is crucial for activation of the RIG-I/MAVS signalling pathway. TRIM25 activity is targeted by influenza A virus non-structural protein 1 (NS1) to suppress IFN production and prevent an efficient host immune response. Here we present structures of the human TRIM25 coiled-coil-PRYSPRY module and of complexes between the TRIM25 coiled-coil domain and NS1. These structures show that binding of NS1 interferes with the correct positioning of the PRYSPRY domain of TRIM25 required for substrate ubiquitination and provide a mechanistic explanation for how NS1 suppresses RIG-I ubiquitination and hence downstream signalling. In contrast, the formation of unanchored K63-linked poly-ubiquitin chains is unchanged by NS1 binding, indicating that RING dimerisation of TRIM25 is not affected by NS1.

[1] Molecular Structure of Cell Signalling Laboratory, The Francis Crick Institute, 1 Midland Road, London NW1 1AT, UK. [2] European Molecular Biology Laboratory, 71 Avenue des Martyrs, 38042 Grenoble, Cedex 9, France. [3] Immunobiology Laboratory, The Francis Crick Institute, 1 Midland Road, London NW1 1AT, UK. [4] Structural and Computational Biology Unit, EMBL Heidelberg, Meyerhofstraße 1, 69117 Heidelberg, Germany. [5] Structural Biology Science Technology Platform, The Francis Crick Institute, 1 Midland Road, London NW1 1AT, UK. Correspondence and requests for materials should be addressed to K.R. (email: katrin.rittinger@crick.ac.uk)

Signalling pathways mediating innate immune responses are regulated by multiple post-translational modifications to allow for a dynamic and specific host response to infection. One of these is ubiquitination, where target proteins are tagged with poly-ubiquitin chains to alter their behaviour and for example change their stability, activity or promote the formation of protein–protein complexes. Ubiquitination is catalysed by E3 ubiquitin ligases that transfer ubiquitin to a lysine side chain of the target and build poly-ubiquitin chains[1]. Tripartite motif (TRIM) family proteins are RING-type E3 ligases that play key regulatory roles in innate immune signalling pathways that lead to the induction of a type I interferon (IFN) response and production of inflammatory cytokines through activation of the IRF3 and NF-κB transcription factors[2,3]. Members of the TRIM family are characterised by a conserved domain architecture that includes an N-terminally located TRIM comprising a catalytic RING domain, one or two B-box domains and a coiled-coil (CC) domain that mediates dimerisation[4–6] (Fig. 1a). The C-terminal

region of TRIMs contains non-catalytic domains, which often recognise E3 ligase substrates and other targets.

Infection of host cells with viral pathogens is sensed by pattern recognition receptors (PRRs) that trigger an antiviral response. RIG-I is a member of the RIG-I-like receptor (RLR) family of PRRs that reside in an auto-inhibited state in uninfected cells in which the N-terminal tandem CARDs (caspase activation and recruitment domains) are unavailable for signalling[7,8]. Activation of RIG-I initiates a signalling cascade that depends on the E3 ligase activity of TRIM25 and induces the production of IFNs and inflammatory cytokines. Interaction with viral RNA and ATP releases the auto-inhibited conformation of RIG-I thereby allowing K63-linked poly-ubiquitination of its second CARD by TRIM25 (Fig. 1a)[8–12]. RIG-I ubiquitination in turn promotes its oligomerization into a helical assembly, enabling interaction with the downstream effector MAVS (mitochondrial antiviral signalling) and formation of filamentous structures that activate a signalling cascade culminating in the production of IFN-α/β[13,14]. Additional K63-linked ubiquitination of RIG-I is mediated by the

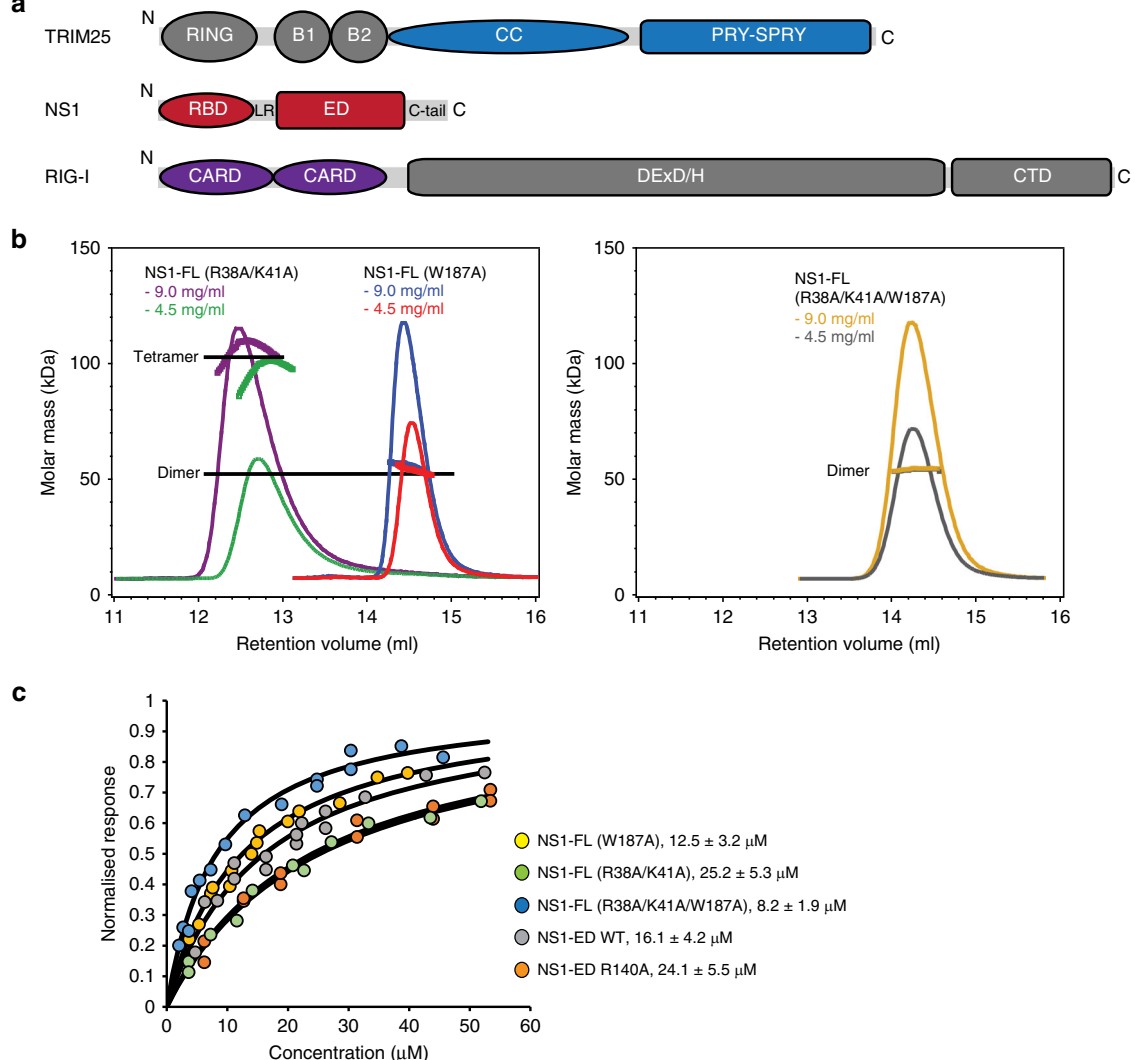

**Fig. 1** Oligomeric state of NS1 and identification of TRIM25-interacting domains. **a** Schematic representation of TRIM25, NS1 and RIG-I domain structures. The crystallised TRIM25 CC-PRYSPRY domain is highlighted in blue and the crystallised RBD and NS1 domains of NS1 in red. The tandem CARD construct of RIG-I used in substrate ubiquitination assays is highlighted in purple. **b** The effect of different mutations on the oligomeric state of NS1-FL assessed by SEC–MALLS. The traces are colour coded according to the NS1 mutant and concentration used. **c** Quantification of the TRIM25-CC interaction with NS1 by biolayer interferometry (BLI). The binding curves are colour-coded and $K_d$s determined are listed. The error is the standard deviation (s.d.) of mean from at least three independent experiments

E3 ligase Riplet and possibly other E3 ligases, though the precise interplay between TRIM25, Riplet and other E3s and their respective contribution to the activation of RIG-I is not fully understood[15–18]. In addition to direct ubiquitination of RIG-I, biochemical experiments have suggested that unanchored K63-linked poly-ubiquitin chains can also activate the RIG-I/MAVS signalling pathway[19]. Structural studies have shown that these chains wrap around the tandem CARDs of RIG-I and form a helical assembly that acts as a seed for MAVS filament formation[20]. Importantly, regardless of the covalent or non-covalent nature of the K63-linked poly-ubiquitin chains involved, the catalytic activity of TRIM25 is crucial for their synthesis and IFN-β promoter activation[21].

Substrate recognition is mediated by the C-terminal PRYSPRY (also known as B30.2) domain of TRIM25, which recognises CARD1 of RIG-I to promote ubiquitination of K172 in CARD2[11,22]. Catalytic activity of TRIM25 requires dimerisation of its RING domains, which may occur in an intramolecular fashion[23], and be enhanced by the PRYSPRY/RIG-I tandem CARD interaction[21]. At present no structures of full-length TRIMs are available and it is not known how the E2-Ub-binding RING domains and the substrate are brought into close proximity to mediate ubiquitin transfer. First insight into the relative position of a PRYSPRY domain with respect to the CC came from the structural characterisation of the TRIM20 CC-PRYSPRY fragment, which showed that the PRYSPRY can adopt multiple positions around the centre portion of the anti-parallel CC dimer[24].

Pathogenic viruses have evolved a variety of mechanisms to inhibit the RIG-I signalling pathway and suppress the host innate immune response[25,26]. An example is non-structural protein 1 (NS1) of influenza A viruses (IAV), a multifunctional virulence factor that binds many host proteins during infection, including TRIM25[27–29]. NS1 is composed of an RNA-binding domain (RBD, a.a. 1–73) followed by a short linker to the effector domain (ED, a.a. 85–202) and an unstructured C-terminal tail (Fig. 1a). Multiple structures of individual domains and two structures of full-length NS1 have provided insight into self-association of NS1, conformational flexibility of the RBD and ED architecture and the interaction with dsRNA and host proteins[28–31]. However, many questions remain, especially regarding the link between the oligomeric state of NS1 and its function. NS1 is a constitutive homodimer, mediated by a high affinity interaction between the RBDs, but forms concentration-dependent higher order oligomers due to weak interactions between EDs via a surface involving W187 and possibly between RBDs[32–34]. Such higher order self-association may be important for some of its functions, such as dsRNA binding[35].

The mechanistic details underlying NS1-mediated inhibition of TRIM25 are not fully understood at present and while an earlier report showed that NS1 binding directly interferes with the E3 ligase activity of TRIM25[27], a recent study suggests that NS1 antagonises binding of TRIM25 to viral ribonucleoprotein particles (vRNPs) thus inhibiting RNA synthesis, an E3 ligase independent activity[36].

To elucidate the mechanism of NS1-mediated suppression of TRIM25 activity we have determined the crystal structures of complexes between the CC of TRIM25 bound to NS1-FL and the isolated ED and characterised their interaction biochemically. To the best of our knowledge, this is the first structure of full-length NS1 bound to a target protein and provides insight into the architecture of NS1-host protein complexes. Our data show that CC-mediated dimerisation of TRIM25 is unperturbed in these complexes, as is its ability to synthesise unanchored K63-linked poly-ubiquitin chains. Instead, a comparison with the structure of a TRIM25 CC-PRYSPRY fragment, also reported here, shows that

NS1 binding displaces the PRYSPRY domain from the CC, thereby preventing RIG-I ubiquitination. Furthermore, our study suggests that the ability of NS1 to self-associate into higher order oligomers and crosslink different molecules of TRIM25, is important for NS1 function.

## Results

**The ED of NS1 forms a stable complex with TRIM25.** The interaction between TRIM25 and NS1 has been suggested to require both the RBD and ED of NS1, while the region in TRIM25 recognising NS1 has been mapped to the CC domain[27] (Fig. 1a). To further characterise complex formation we aimed to quantitate the interaction using purified full-length NS1 (NS1-FL) of strain A/Puerto Rico/8/1934 H1N1 (hereafter referred to as NS1) and the coiled-coil domain of TRIM25 (TRIM25-CC)[27]. Unfortunately, wild-type NS1-FL aggregates even at low concentrations. Consequently, past structural studies were carried out on constructs containing two mutations (R38A/K41A) that largely suppress aggregation (Fig. 1b) and this allowed structure determination of full-length NS1[29,30]. However, these mutations were reported to prevent interaction with TRIM25-CC[27]. Therefore, we attempted to identify novel NS1-FL constructs with improved solubility. Consistent with previous observations[32], mutation of W187A in the ED greatly improves protein solubility of NS1 and SEC–MALLS analysis shows it is mainly dimeric with some minor propensity to form concentration-dependent oligomers (Fig. 1b). In contrast, a triple mutant protein (R38A/K41A/W187A) is strictly dimeric in solution (Fig. 1b). The ability of these proteins to interact with TRIM25 was investigated by bio-layer interferometry (BLI). NS1 W187A bound TRIM25-CC with an affinity of 12.5 μM, which was not reduced in the triple mutant (8.2 μM) (Fig.1c). In light of this observation, we tested binding of individual NS1-RBD and NS1-ED to TRIM25-CC. These experiments showed that the ED alone binds TRIM25-CC with an affinity of 16.1 μM, whereas no binding could be detected with the RBD alone at concentrations up to 300 μM (Fig. 1c). In conclusion, these experiments establish that the effector domain of NS1 is sufficient to form a stable complex with the coiled-coil domain of TRIM25.

**Structure of the core NS1-ED/TRIM25-CC complex.** To gain molecular insight into the interaction between NS1 and TRIM25 we determined the crystal structure of the NS1-ED/TRIM25-CC complex at 2.8 Å resolution by molecular replacement ($R/R_{\text{free}} = 0.20/0.22$) (Table 1 and Supplementary Fig.1a). The complex crystallised in space group C222 with three copies of each NS1-ED and TRIM25-CC monomer in the asymmetric unit (Fig. 2a). Symmetry analysis shows that each NS1-ED binds to a TRIM25-CC dimer via two separate interfaces, interface A and interface B, which bury 872 and 611 Å[2], respectively, of solvent accessible surface (Fig. 2b). Complex formation between TRIM25-CC and NS1-ED does not induce major conformational changes in either protein and the structures of the bound and unbound domains overlap with RMSD values of 2.41 Å (for 156 residues) for TRIM25-CC (PDB:4LTB) and 0.56 Å (for 117 residues) for NS1-ED and apo-NS1-ED (PDB:3O9T) (Supplementary Fig. 1b–d)[32,37]. The biggest difference between apo TRIM25-CC and the NS1-bound state is seen in the linker connecting helix α2 to the central helix α3, which is pushed from its position in the apo TRIM25-CC structure towards helix α1 due to contacts by NS1-ED from interface A. This copy of NS1-ED makes contacts with both TRIM25 chains within the CC dimer via a highly conserved motif of NS1 that includes a short α helix comprising residues 95–99 (Fig. 2c). Previous mutational analysis identified E96 and E97 in this motif as crucial for TRIM25 binding and

**Table 1 Data collection and refinement statistics**

|  | TRIM25CC/ NS1-ED (PDB: 5NT1) | TRIM25CC/ NS1-FL (PDB: 5NT2) | TRIM25CC-PRYSPRY (PDB: 6FLN) |
|---|---|---|---|
| Data collection |  |  |  |
| Space group | $C\,2\,2\,2$ | $P\,1$ | $P\,6_{1}22$ |
| Cell dimensions |  |  |  |
| $a, b, c$ (Å) | 131.38, 225.23, 146.43 | 73.26, 76.30, 92.07 | 89.92, 89.92, 827.09 |
| $\alpha, \beta, \gamma$ (°) | 90.0, 90.0, 90.0 | 100.04, 93.71, 111.14 | 90.0, 90.0, 120.0 |
| Resolution (Å) | 48.68–2.82 (2.89–2.82)[a] | 61.51–4.26 (4.33–4.26)[a] | 50–3.60 (3.69–3.60)[a] |
| $R_{merge}$ | 0.092 (1.29)[a] | 0.19 (0.6)[a] | 0.14 (2.4)[a] |
| $I/\sigma I$ | 14.9 (1.4)[a] | 5.7 (2.5)[a] | 8.2 (0.8)[a] |
| Completeness (%) | 99.9 (100.0)[a] | 98.6 (97.1)[a] | 99.5 (98.5)[a] |
| Redundancy | 6.8 (5.7)[a] | 2.6 (2.6)[a] | 5.4 (5.5)[a] |
| Refinement |  |  |  |
| Resolution (Å) | 48.68–2.82 | 61.68–4.26 | 50–3.60 |
| No. reflections | 355,400 | 32,644 | 131,681 |
| $R_{work}/\,R_{free}$ | 0.195/0.224 | 0.272/0.309 | 0.287/0.318 |
| No. atoms |  |  |  |
| Protein | 7030 | 20,802 | 8861 |
| Water | 8 | 0 | 0 |
| B-factors |  |  |  |
| Protein | 79.3 | 122.2 | 225.2 |
| R.m.s deviations |  |  |  |
| Bond lengths (Å) | 0.006 | 0.004 | 0.008 |
| Bond angles (°) | 0.8 | 0.8 | 1.05 |

One crystal was used for each structure
[a]Values in parentheses are for highest-resolution shell

inhibition of interferon production[27]. Our structure now shows that the side chains of E96 and E97 point away from the binding interface with TRIM25. However, they form a network of contacts within NS1-ED, especially between the side chains of E96 and R100, which is likely important to maintain the structural integrity of this helix (Fig. 2c). Instead, $L95_{ED}$ of this helix is a key element of the interface and contacts a hydrophobic pocket of TRIM25 formed by residues from both chains of the CC including $V223_{CC}$, $F274_{CC}$, $I277_{CC}$, $I324_{CC}$ and $V327_{CC}$. This interaction is further stabilised by a hydrogen bond between the backbone of $L95_{ED}$ and the side chain of $E326_{CC}$. Additional interactions between NS1-ED and one of the CC monomers involve $Y89_{ED}$, which forms hydrogen bonds with $D222_{CC}$ and $D229_{CC}$, and between $E101_{ED}$ and $Q212_{CC}$. (Fig. 2c). Interface B involves loop-forming NS1-ED residues 139-143 that contact the central four-helical bundle of TRIM25-CC with $R140_{ED}$ at its centre (Fig. 2d). To investigate the relative importance of either interface in solution we measured binding of ED constructs containing mutations $L95A_{ED}/S99A_{ED}$ or $R140A_{ED}$ to TRIM25-CC by BLI. These experiments showed that $R140A_{ED}$ still binds with 24.1 μM affinity (Fig. 1c, orange circles), whereas almost no instrument response was detected up to 125 μM of the ED mutant $L95A_{ED}/S99A_{ED}$, indicating that the mutations significantly weaken the interaction. We conclude that interface A is the most relevant for the direct interaction of NS1-ED with TRIM25 CC.

**Structure of NS1-FL in complex with TRIM25-CC.** While our experiments established that the effector domain of NS1 is sufficient to form a stable complex with TRIM25, we wondered how the constraints imposed by the constitutively dimeric RBD within full-length NS1 might influence the architecture of the NS1/

TRIM25 assembly. To gain insight into this arrangement we crystallised a complex between TRIM25-CC and the triple mutant NS1-FL (R38A/K41A/W187A) and solved the structure by molecular replacement at 4.26 Å resolution ($R/R_{free} = 0.27/0.31$) (Table 1). Electron density is observed for two TRIM25 CC dimers, one complete NS1-FL dimer and two additional NS1-ED domains (Fig. 3a; Supplementary Fig. 2a).

This structure is the first complex between full-length NS1 and a target protein, and provides crucial insight into the spatial arrangements of the RBD and ED during target binding. The RBDs of the NS1 dimer form the canonical homodimeric 6-helical bundle as observed in the structures of unbound NS1-FL (PDB: 3F5T and 4OPH), the RBD alone (PDB: 1AIL) and an RBD/dsRNA complex (PDB: 2ZKO)[29,30,35,38]. Interestingly, the two NS1 monomers within the homodimer are not symmetrical; while both monomers align well on the NS1-RBD (RMSD of 0.6 Å), one NS1-ED is shifted towards the stem in relation to the other by ~17 Å (Fig. 3b, c). The relative arrangement of RBD and ED seen in this structure is different from previous structures of NS1-FL, supporting the model that the linker region acts as a flexible hinge to allow movement of the ED relative to the RBD to accommodate binding of different substrates (Fig. 3d)[29].

The only contacts between TRIM25-CC and NS1-FL are made by the ED, in agreement with our BLI experiments. Structural superposition of the NS1-ED/TRIM25-CC complex with that of NS1-FL/TRIM25-CC shows that interface A, but not interface B, is present in the full-length complex (RMSD 0.7 Å) (Supplementary Fig. 2b), confirming its physiological relevance. Our two structures of NS1/TRIM25-CC complexes clarify the oligomeric state of TRIM25 upon binding NS1. Previous studies had suggested that NS1 interferes with CC-mediated dimerisation of TRIM25 and thereby suppresses activity[27,39]. However, dimerisation is unperturbed in both TRIM25/NS1 complexes and instead the dimeric CC is recognised by two NS1 effector domains at its distal ends in a symmetrical fashion (Figs. 2 and 3). Furthermore, a key observation from the NS1-FL/TRIM25-CC structure is that although the overall architecture of the complex is identical to that of NS1-ED/TRIM25-CC, the two EDs of a given NS1 homodimer bridge two different TRIM25 dimers that are located in a perpendicular arrangement in the crystal (Fig. 3a). We hypothesise that this arrangement highlights the critical role of the constitutive NS1-RBD dimer in mediating higher order oligomerization of the NS1–TRIM25 complex, which is likely important for the proviral function of NS1. These results raise the question as to how complex formation with NS1 interferes with the ability of TRIM25 to ubiquitinate RIG-I. To investigate this further, we aimed to gain further structural insight into the functional architecture of TRIM25 by determining the position of the substrate-binding PRYSPRY domain.

**The structure of TRIM25 CC-PRYSPRY.** The structure of a human TRIM25 CC-PRYSPRY construct (189–630) was determined at 3.6 Å resolution (Table 1). The structure was solved by molecular replacement using the CC structure (PDB:4LTB) and a de novo determined structure of the human PRYSPRY domain (residues 434–630) at 2.0 Å resolution (Supplementary Table 1). The human PRYSPRY domain structure is very similar to that of mouse TRIM25 (residues 440–634, PDB:4B8E)[22] with an RMSD of 0.64 Å for 189 matched Cα atoms and sequence identity of 81.5% (Supplementary Fig. 3). The CC-PRYSPRY structure has one and a half CC dimers in the asymmetric unit with a crystallographic twofold completing the second dimer (Supplementary Fig. 4a–c). Each bow-shaped CC dimer has two PRYSPRY domains bound towards the ends, maintaining the twofold

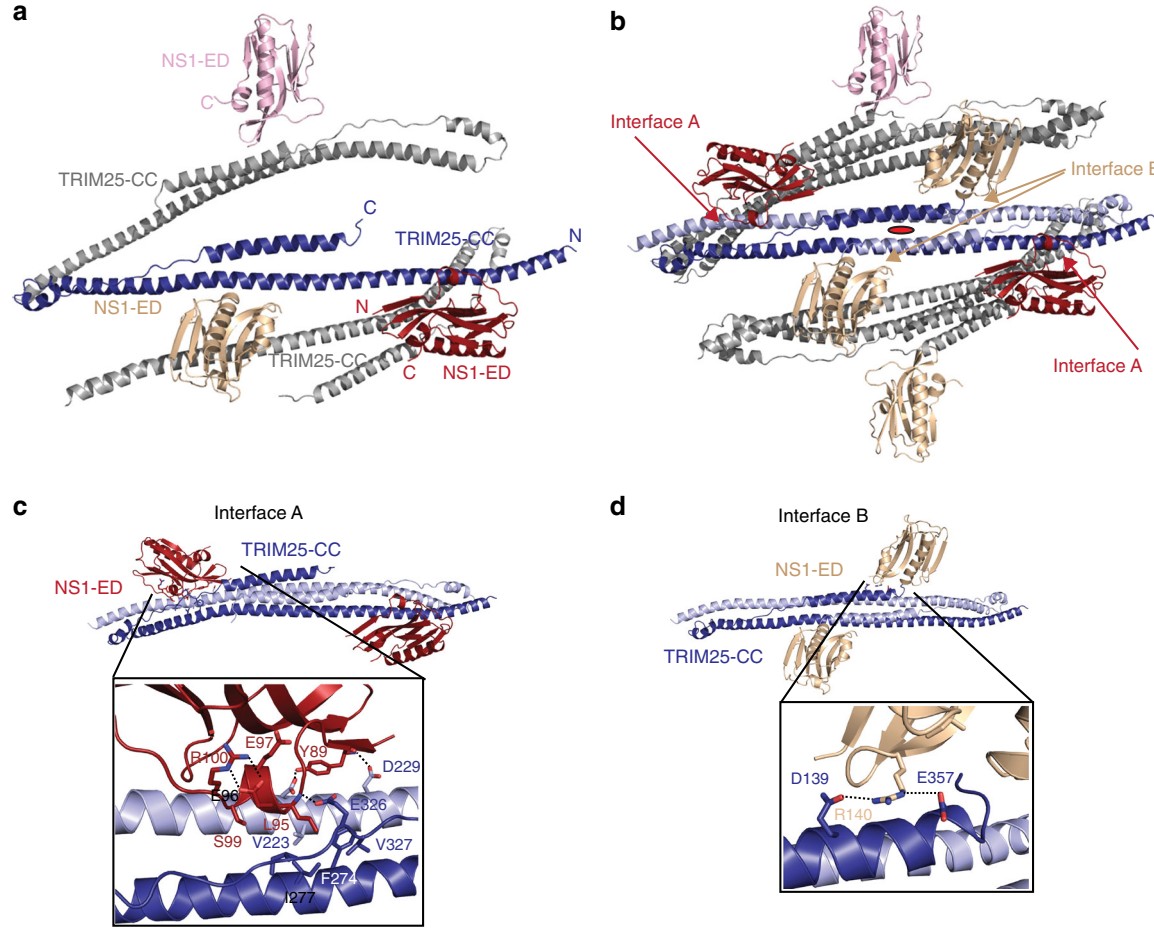

**Fig. 2** Crystal structure of NS1-ED in complex with TRIM25-CC. **a** The asymmetric unit of the TRIM25-CC/NS1-ED complex containing three copies of each TRIM25-CC (blue and grey) and NS1-ED (in red, wheat and pink) monomers. **b** Symmetry analysis shows that NS1-ED binds to a dimeric TRIM25-CC via two distinct interfaces. The structure is shown in ribbon representation. The position of a twofold symmetry axis is shown in red. **c** Interface A is formed by the NS1-ED α-helix containing amino acids E95 and E96, which have previously been suggested to be important for complex formation. The insert shows a close-up view of the interface with TRIM25-CC in light and dark blue and NS1-ED in red. Interactions described in the text are highlighted by dashed lines. **d** Details of interface B, with TRIM25-CC in light and dark blue and NS1-ED in wheat

symmetry (Fig. 4a). The CC structure with the PRYSPRY is only slightly distorted from that of the CC alone (PDB:4LTB, RMSD 1.16 Å for 169 aligned Cα). The 73-residue long linker (residues 361–433) connecting the CC and PRYSPRY domain is not visible in the electron density so the exact connectivity is uncertain (Fig. 4a). The footprint of the PRYSPRY domain on the CC, which corresponds to a total buried surface area of 1433 Å$^2$, spans residues 269–287 on helix α1 and residues 320–326 on the extended connection between α2 and α3, all on the same monomer (Fig. 4b, Supplementary Fig. 3). These regions contact PRYSPRY domain surface loops spanning residues 460–464 (L1), 472–476 (L2), 488–494 (L3) and 504–506 (L4) (Fig. 4b, Supplementary Fig. 3). Of note, there are four tyrosines at the interface (Tyr323, Tyr463, Tyr476 and Tyr488) which are highly conserved as tyrosine or phenylalanine in mammal and bird TRIM25 (Supplementary Fig. 3).

To gain insight into the dynamics of the TRIM25 CC-PRYSPRY domain arrangement we performed small-angle X-ray scattering on the CC-PRYSPRY construct (Supplementary Fig. 5a, b). In parallel we performed NMR titrations between the separate CC and PRYSPRY domains (Supplementary Fig. 5c–g). These experiments show that the interaction between the two domains is weak in solution but involves the interface observed in the crystal structure as mutation of Y463S and Y476S weakened the interaction even further (Supplementary Fig. 5g). The functional

importance of this interface for the correct positioning of the PRYSPRY to enable substrate ubiquitination is highlighted by the almost complete loss of RIG-I-2CARD ubiquitination upon mutation of these residues (Fig. 4c) although the overall fold of the PRYSPRY domain remains intact (Supplementary Fig. 5f). Taken together, these experiments suggest that in the absence of other components of the system, the PRYSPRY domain is in a dynamic equilibrium between a bound state and a constrained diffusing state tethered to the CC by the flexible linker, in agreement with recent results[40], but that the bound state is important to enable substrate ubiquitination.

Superposition of the NS1-ED and PRYSPRY-bound CC structures shows that the two domains bind at the same position near the end of the CC but on opposite sides such that they would sterically clash only minimally (Fig. 4d). NS1-ED binds mainly to the region spanning residues 212–229 of helix α1 of one monomer but also to residues 274–277 of helix α1 from the other monomer. The PRYSPRY domain only binds to helix α1 of the second monomer (residues 269–287) (Supplementary Fig. 3). However, a key observation is that both domains bind to the extended region between helices α2 and α3 (residues 316–331) and strikingly, binding of NS1-ED necessitates a considerable distortion of this region away from its structure in the PRYSPRY-bound or PRYSPRY-unbound CC (Fig. 4d). Thus, while the binding sites of the PRYSPRY domain and NS1-ED are on

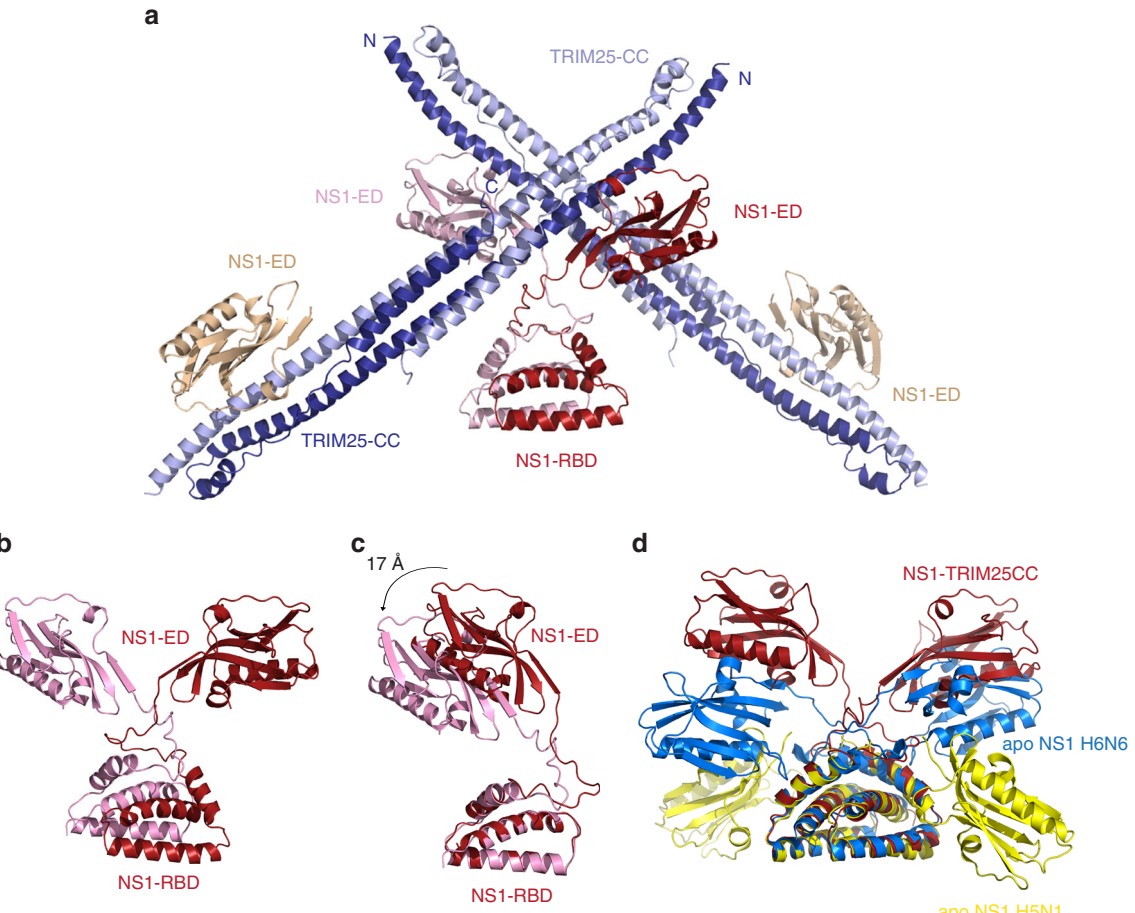

**Fig. 3** Crystal structure of NS1-FL in complex with TRIM25-CC. **a** The asymmetric unit contains four molecules of each, NS1-FL (R38A/K41A/W187A) and TRIM25-CC, with electron density visible for 2 TRIM25-CC dimers (in dark and light blue), 2 NS1-FL (in red and pink) and 2 NS1-ED domains (in wheat). **b** NS1-FL resembles the shape of letter Y, with the RBD forming a homodimeric 6-helical structure and the effector domains extending outwards without making contact with each other or the RBDs. **c** Structural alignment of dimeric NS1-FL from **a** using the RBD for the overlap shows that the two protomers are not in identical positions, with one ED being shifted by 17 Å in relation to the other. **d** Structural alignment of NS1 shown in **b** (in red) with the other two available apo-NS1-FL structures (A/Vietnam/1203/2004(H5N1), R38A/K41A) shown in yellow (PDB: 3F5T) and (A/MN/993/1980(H6N6) R38A/K41A) shown in blue (PDB: 4OPH) reveals structural flexibility owing to the linker region (LR) of NS1

opposite sides of the CC, both contact the α2 and α3 linker and simultaneous binding is impossible.

**Inhibition of TRIM25 activity by NS1.** To test if the changes brought upon by NS1 binding interfere with the correct juxtaposition of the E2~ubiquitin-binding RING and substrate-binding PRYSPRY domains and hence substrate ubiquitination, we analysed TRIM25-mediated RIG-I ubiquitination in cells. In the absence of NS1, overexpression of TRIM25 FL induced a significant increase in RIG-I-2CARD ubiquitination, which was severely suppressed in the presence of NS1 WT (Fig. 5a). Mutation of R38A, K41A or W187A in NS1 did not affect the ability of NS1 to suppress RIG-I ubiquitination, in accordance with our binding studies (Figs. 1c and 5a). Interestingly, combining mutation Y89A, with mutation of L95A/S99A, which severely weakened complex formation between TRIM25 and the isolated ED, had only a minor effect on the ability of NS1-FL to suppress RIG-I ubiquitination (Figs. 2c and 5a). This observation strongly suggests that FL NS1-mediated higher order oligomerization of TRIM25, and potentially higher order self-association of NS1, are important factors in the mechanism of suppression and can overcome a weakened direct NS1–TRIM25 interaction due to avidity effects.

Given that NS1 does not interfere with CC-mediated dimerisation of TRIM25 and instead prevents the correct positioning of the PRYSPRY domain on the CC required for substrate ubiquitination, we wondered what the effect of NS1 might be on free K63-linked poly-ubiquitin chain synthesis. We have previously shown that the RING domains of TRIM25 need to be dimeric for catalytic activity and have proposed a model in which RING dimerisation may occur in an intramolecular fashion[23]. To examine whether NS1 binding to TRIM25 affects catalytic activity we performed UbcH5C~Ub discharge assays with recombinant TRIM25-FL in the presence of increasing concentrations of different NS1 constructs to assess the substrate-independent E3 ligase activity of TRIM25 (Supplementary Fig. 6). These experiments show that TRIM25-mediated discharge of UbcH5C~Ub remains unaltered in the presence of all NS1 proteins tested, suggesting that NS1 binding does not interfere with the catalytic activity of TRIM25 per se. To further validate this observation, we performed ubiquitination assays with heterodimeric UBE2N/UBE2V1 that catalyses the formation of unanchored K63-linked poly-ubiquitin chains in the absence of RIG-I (Fig. 5b and Supplementary Fig. 6c). These experiments clearly show that poly-ubiquitin chain formation is also unaffected by the presence of NS1. Taken together, these experiments demonstrate that the intrinsic catalytic activity of

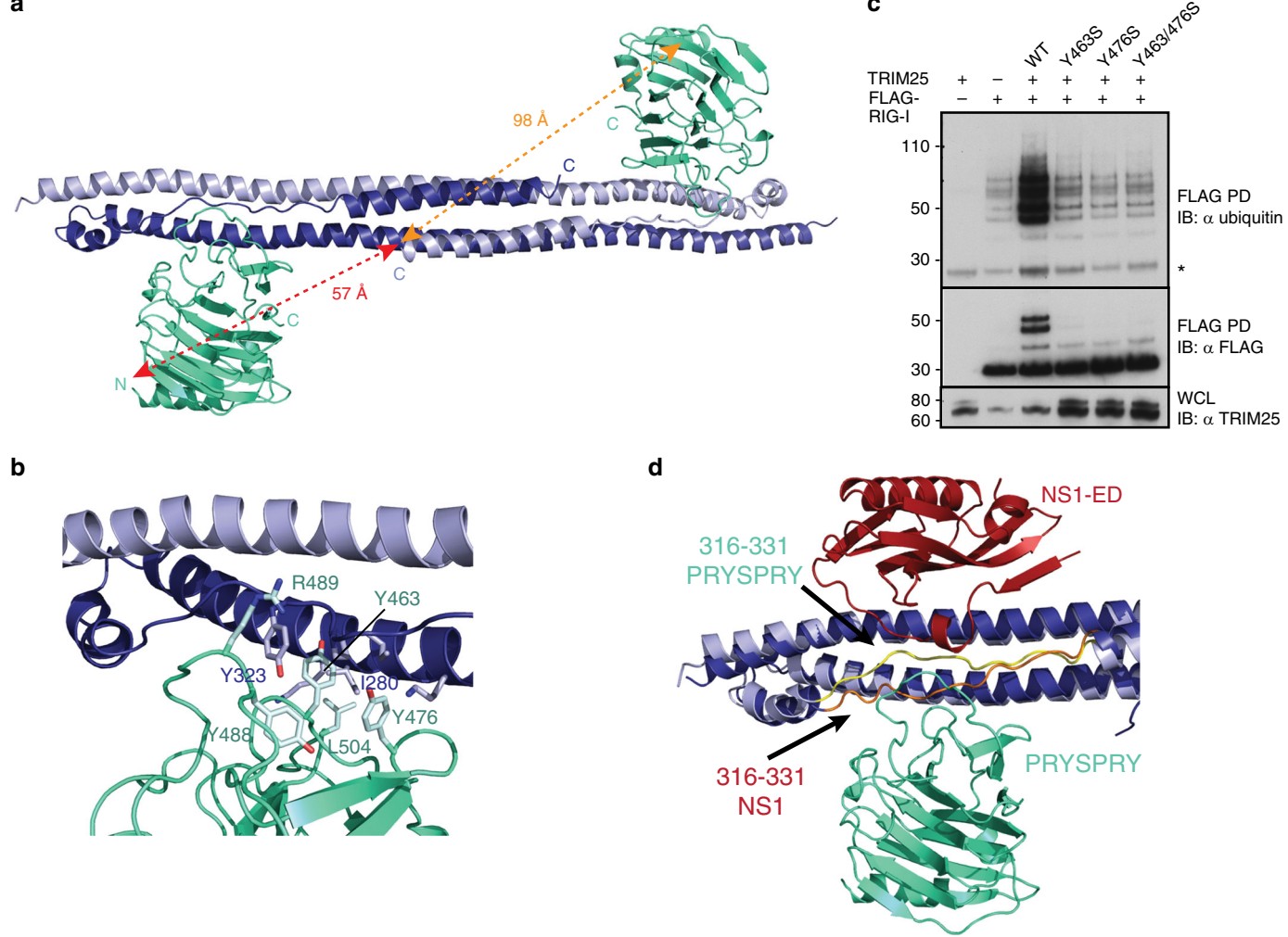

**Fig. 4** Crystal structure of TRIM25 CC-PRYSPRY. **a** The TRIM25 CC dimer (in dark and light blue) with two PRYSPRY domains bound at either end in green cyan. There is no electron density for the linker connecting the CC and PRYSPRY domains. The distance between the N-termini of the PRYSPRY domains and C-terminus of one CC chain is indicated. **b** Close-up view of the interface between the CC and PRYSPRY domain. Residues described in the text are shown as ball and stick. **c** Ubiquitination assays of the tandem CARDs of RIG-I with TRIM25 to test the importance of Y463 and Y476 in stabilising the observed CC-PRYSPRY arrangement and promoting substrate ubiquitination. FLAG-RIG-I-2CARD and TRIM25 were co-transfected into HEK293T cells and WCLs were subjected to IP with anti-FLAG beads and immunoblotted with α-ubiquitin and α-FLAG antibodies. The asterisk indicates an unspecific band. **d** Comparison of the positions of the PRYSPRY domain (green cyan) and the NS1 ED (red), highlighting the changes in the position of the linker connecting the CC and PRYSPRY domain, induced by binding to NS1

TRIM25 is not suppressed even in the presence of high concentrations of NS1.

Next, we tested how the loss of RIG-I ubiquitination affected signalling downstream of RIG-I (Fig. 5c). Expression of the tandem CARDs of RIG-I is sufficient to induce spontaneous activation of the RIG-I/MAVS pathway, even in the absence of an RNA ligand[11]. Indeed, expression of RIG-I-2CARD, in combination with TRIM25, triggered type I IFN production, as measured using a reporter plasmid expressing the firefly luciferase gene under the control of the IFN-β promoter (Fig. 5c). WT NS1 potently suppressed RIG-I-2CARD-induced expression of the IFN-β promoter-regulated luciferase reporter, as shown previously[27] (Fig. 5c). This suppression activity was not affected by mutations R38A, K41A or W187A in accordance with our binding and ubiquitination assays. Similarly, the L95A/S99A/Y89A mutant, which only had a minor effect on RIG-I ubiquitination in FL NS1 (Fig. 5a), is still capable of inhibiting the interferon response. This further indicates that even a low affinity direct interaction between NS1 and TRIM25 is sufficient

for suppression due to avidity effects through higher order oligomerization. In summary, our structures and ubiquitination assays show that NS1 binding to TRIM25 inhibits RIG-I ubiquitination by steric effects, while leaving the ability of TRIM25 to form unattached K63 chains intact.

## Discussion
RIG-I is an intracellular PRR that plays a key role in the innate immune response against a variety of viruses. Recognition of viral RNA by RIG-I initiates a signalling cascade that culminates in the production of IFNs and proinflammatory cytokines. Ubiquitination of RIG-I by TRIM25 is crucial for the activation of the RIG-I/MAVS signalling pathway establishing TRIM25 as a key player in antiviral immunity. Conversely, viruses have evolved intricate mechanisms to interfere with the host immune response, one of which is the sequestration of TRIM25 by influenza A virus NS1. This interaction was originally suggested to suppress RIG-I ubiquitination and IFN production, although a recent study

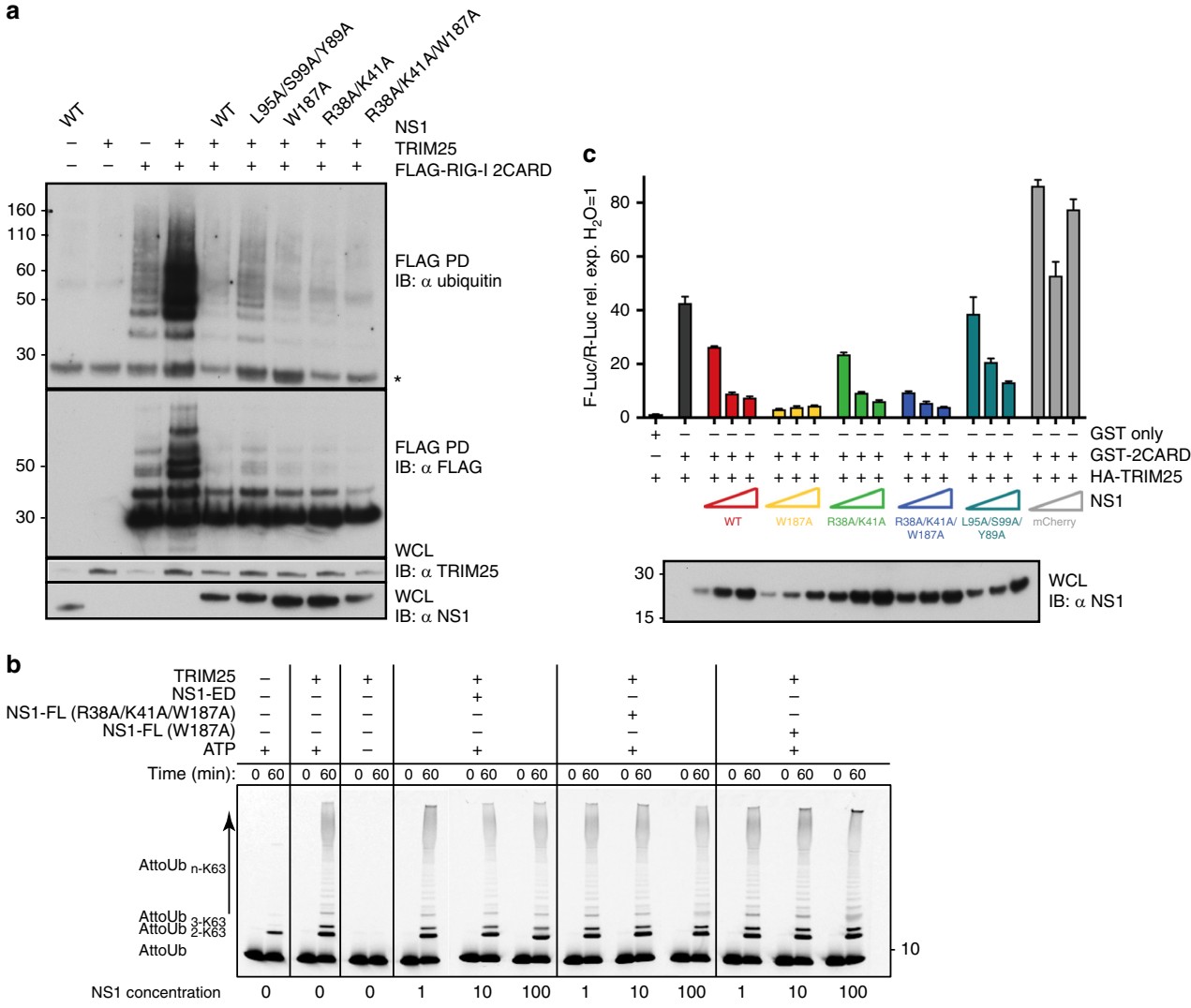

**Fig. 5** Inhibition of TRIM25-mediated RIG-I ubiquitination and signalling by NS1. **a** Effect of NS1 and mutants on the ubiquitination of 3FLAG-RIG-I-2CARD by TRIM25 in HEK293T cells. 3FLAG-RIG-I-2CARD, TRIM25 and NS1 were co-transfected into HEK293T cells and WCLs were subjected to IP with anti-FLAG beads and immunoblotted with α-ubiquitin and α-FLAG antibodies. The asterisk indicates an unspecific band. **b** In vitro ubiquitination assays with UBE2N/UBE2V1 and TRIM25-FL in the absence or presence of increasing amounts of NS1 supplemented with fluorescent AttoUb. Assays were carried out with NS1-ED or NS1-FL (R38A/K41A/W187A or W187A) and the reaction was monitored over 60 min. Gels were scanned with a Typhoon FLA 9500 scanner and the fluorescence converted to black and white. **c** 3FLAG-RIG-I-2CARD, TRIM25, and increasing amounts of NS1 WT and mutants (25, 50 and 100 ng) were co-transfected into HEK293T cells. Interferon induction by 3FLAG-RIG-I-2CARD was measured in a Luciferase-based reporter assay, where Luciferase activity is controlled by the IFN-β promoter. The mCherry plasmid was used as a control. The error bars shown are the standard deviation (s.d.) of mean from two independent experiments, each with triplicates

proposed instead that the physiologically important consequence of TRIM25/NS1 complex formation was prevention of vRNP-binding by TRIM25[27,36]. Both the RBD and ED of NS1 were suggested to be required for TRIM25 recognition and suppression of its E3 ligase activity[27]. Here we show by quantitative analysis of the TRIM25/NS1 interaction that the ED is sufficient to form a stable complex (Fig. 1c). To understand NS1-mediated inhibition of TRIM25 activity on a molecular level we have determined crystal structures of the CC domain of TRIM25 in complex with NS1-FL and the isolated ED, and of a TRIM25 CC-PRYSPRY fragment, which includes the RIG-I-recognition domain of TRIM25. These structures establish that NS1 binding does not inhibit TRIM25 function by interfering with the CC-mediated dimerisation of TRIM25. Instead, two NS1 molecules bind symmetrically to each monomer of the dimeric CC (Figs. 2 and 3). Moreover, a structural comparison between the TRIM25/NS1

complex and the TRIM25 CC-PRYSPRY fragment reveals that NS1 and the PRYSPRY domain bind in a mutually exclusive fashion to overlapping sites on the CC (Fig. 4d). Using interface disrupting mutations we show that PRYSPRY domain binding to the CC is required for RIG-I CARD ubiquitination (Fig. 4c). Thus, NS1 competition for the CC provides a mechanism for NS1 interference with RIG-I ubiquitination and downstream signalling. It is unlikely that NS1 binding interferes with substrate recognition per se: the PRYSPRY domain is not contacted by NS1 and it has previously been shown that NS1 does not inhibit TRIM25/RIG-I complex formation[27,39]. Instead, our data are consistent with a model in which TRIM25/NS1 complex formation prevents the correct functional juxtaposition of the RING and PRYSPRY domains required to bring the substrate into close proximity with the E2~Ub conjugate to promote ubiquitin transfer (Fig. 6).

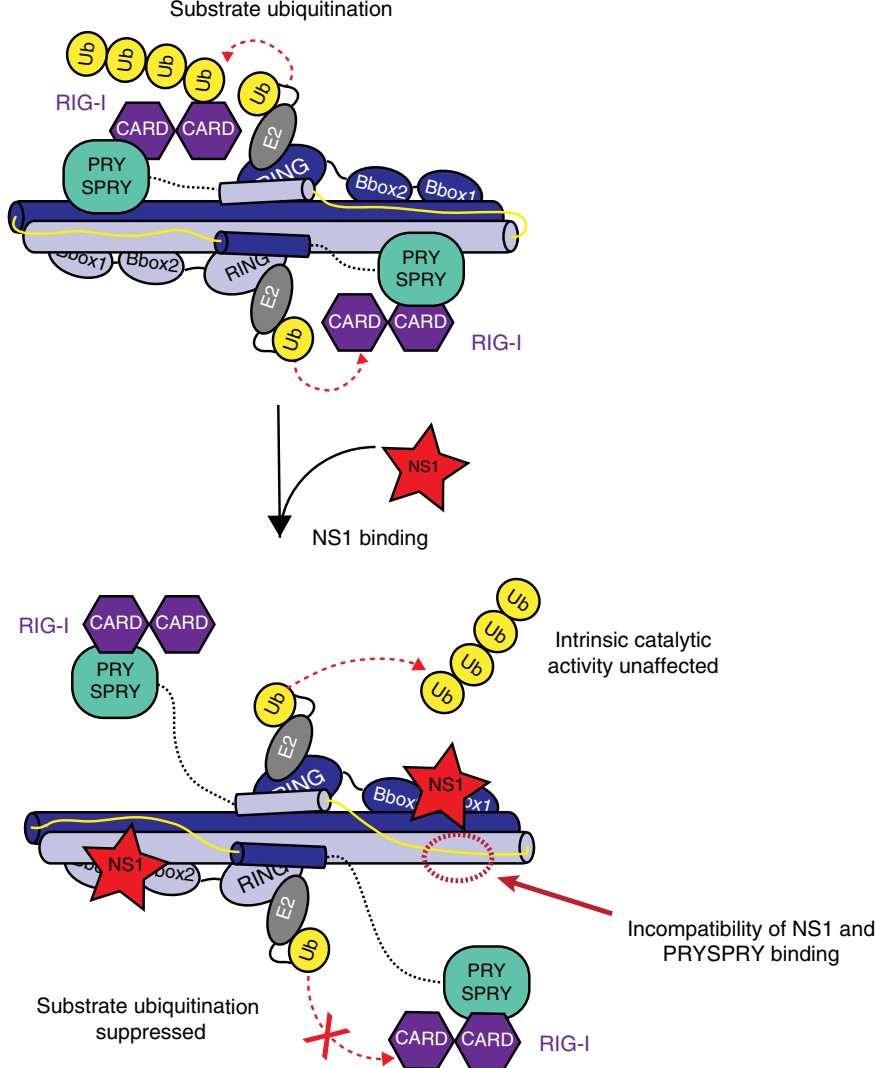

**Fig. 6** Model for suppression of substrate ubiquitination by NS1. Model of TRIM25 activity highlighting how the RING domains either side of the anti-parallel CC bind back to dimerise in an intramolecular fashion and come into close proximity of the substrate-binding PRYSPRY domain. The monomers of dimeric TRIM25 are shown in light and dark blue, with ubiquitin-loaded E2 (E2-Ub) in grey and yellow bound to each RING domain while the tandem CARDs of RIG-I (in purple) are bound to the PRYSPRY domain (in green cyan) of TRIM25. The linker connecting the CC and PRYSPRY domain is highlighted in yellow. Binding of NS1 (in red) to the CC of TRIM25 interferes with the linker position and hence correct arrangement of the PRYSPRY domain with respect to the RING domains to suppress substrate ubiquitination

Intriguingly, structure-based mutations in NS1 (L95A/S99A/Y89A) that severely weaken the interaction between the isolated NS1-ED and TRIM25-CC in vitro still suppress RIG-I ubiquitination in cells (Fig. 5a). In conjunction with our observation that NS1-FL connects two different TRIM25 dimers and the natural tendency of wild-type NS1 to form higher order oligomers and aggregates, this suggests that a productive TRIM25–NS1 interaction requires two components: a direct protein–protein interface to provide specificity as determined in our structures, and the propensity of NS1 to induce the formation of large TRIM25/NS1 assemblies. In accordance with the ubiquitination assays, all mutants significantly suppressed RIG-I-induced IFN expression. This includes the NS1 R38A/K41A mutation located in the RBD, which was previously suggested to abrogate the inhibitory effect of NS1 on RIG-I signalling[27]. This mutation has been shown to interfere with RNA-binding, which is required for NS1 function during influenza virus infection[41,42]. However, our structures show that these residues do not contact TRIM25 directly, consistent with our observation that they do not affect ubiquitination

of the tandem CARDs of RIG-I or IFN production in cells. Nevertheless, the R38A/K41A mutation suppresses NS1 self-association, at least partially (Fig. 1b)[29,30], and we speculate that depending on experimental conditions this may result in the apparent loss of RIG-I ubiquitination as observed previously[27]. Further studies will be required to fully understand the precise role of NS1 self-association in the inhibition of RIG-I ubiquitination by TRIM25 and the suppression of a host immune response. Interestingly, although binding of NS1 strongly suppresses substrate ubiquitination, it has no effect on the ability of TRIM25 to synthesise unanchored K63-linked poly-ubiquitin chains. Given that overexpression of NS1 suppresses IFN production in luciferase reporter assays, this implies that unanchored K63 chains are not sufficient to activate the RIG-I/MAVS pathway under these conditions.

Comparison of our NS1/TRIM25 structure with the NS1-ED/p85β structure (PDB: 3L4Q) reveals remarkable similarities in the manner in which the NS1-ED recognises its targets[43,44]. p85β, a regulatory subunit of the phosphoinositide 3-kinase (PI3K)

complex, is a multi-domain protein containing a rigid elongated CC domain (β-iSH2) which is targeted by IAV NS1 to stimulate PI3K activity. Intriguingly, NS1-ED binds β-iSH2 through the same interface observed in our complex with TRIM25-CC, involving the short NS1 α-helix comprising residues 95–99 that packs against the target CC region (Supplementary Fig.7). The fact that NS1 uses the same surface to bind similar structural features in diverse target proteins suggests one mechanism by which viral proteins can evolve multi-functionality. However, the overlap in binding surfaces used to target different host proteins also implies that the analysis of NS1 mutations in recombinant virus assays is more complicated than previously assumed as a given mutation may affect interaction of NS1 with multiple targets and hence affect multiple host activities. More NS1/host protein structures are required to fully appreciate how frequently viral proteins use overlapping binding surfaces to target different host proteins.

At present, no structure of a full-length TRIM protein is available that could provide molecular insight into the overall arrangement of the E3/E2-Ub conjugate/substrate complex during the ubiquitin transfer reaction. However, structural characterisation of the CC domains of a number of TRIM family proteins have shown that they all form anti-parallel dimers of ~170 Å in length, which based on sequence conservation likely constitutes a general property of this protein family[37] (Supplementary Fig. 8). The crystal structure of a CC-PRYSPRY fragment of TRIM20 shows that the two PRYSPRY domains are positioned close to the dimer twofold axis with a centre of mass separation of only ~45 Å (PDB:4CG4). This is a consequence of the short helical linker between the end of the helical CC and the PRYSPRY domain in TRIM20 (Supplementary Fig. 8b). Even so, the linker can bend leading to some variability in the position of the PRYSPRY domain, which does not itself make any stabilising contact with the CC. In contrast, the TRIM25 linker is much longer, although of variable length and sequence between species, and is unstructured, allowing the PRYSPRY domain to form a transient interaction nearer the CC ends (centre of mass separation ~110 Å) that is important for substrate ubiquitination (Fig. 4a). About half of all TRIM family proteins contain a PRYSPRY domain, each adapted to recognise a different substrate. The observed variety in linker flexibility and PRYSPRY position may be important to accommodate substrates of different sizes and position them in the correct orientation to facilitate their ubiquitination. Although no structure of the TRIM25 tandem B-boxes exists, the structure of the TRIM5α CC with its single B-box indicates their probable location (Supplementary Fig. 8c). Finally, superposition of the TRIM25 CC-PRYSPRY structure with the TRIM21 PRYSPRY-Fc structure suggests that the equivalent, putative substrate-binding surface of the TRIM25 PRYSPRY domain would remain accessible when the PRYSPRY domain is bound to the CC (Supplementary Fig. 8d), although recently it was proposed that a cryptic second site on the PRYSPRY domain would also bind the RIG-I CARDs[40].

In this context it is interesting to note that a number of proteins have been identified that interact with the CC region to regulate TRIM protein activity. Some of these are viral proteins that suppress TRIM activity in order to interfere with an antiviral response. Examples are IAV NS1 as described in this study or the immediate-early protein IE1 from herpes virus human cytomegalovirus (HCMV), which antagonises the antiviral activity of TRIM19/PML nuclear bodies[45,46]. Others like MAGE (melanoma antigen genes) proteins have been shown to bind the CC domain of TRIM28 to enhance ligase activity and possibly contribute to substrate targeting[47,48]. Based on these observations we hypothesise that the CC domain of TRIM ligases might act as a more

general protein interaction platform that is exploited by pathogens as well as oncogene products to regulate ligase activity and/or redirect substrate specificity.

## Methods

**Protein production and purification.** Cloning, expression and purification of His-Ube1, UBE2D3 (UbcH5c), UBE2N (Ubc13), UBE2V1 (UEV1) and M1C-ubiquitin have been described before[49,50].

NS1 (strain A/Puerto Rico/8/1934, UniProt P03496) constructs NS1-RBD (1–73), NS1-ED (79–230) and NS1 mutants NS1-FL (W187A), NS1-FL (R38A/K41A/W187A), NS1-FL (R38A/K41A), NS1-ED (L95A/S99A), NS1-ED (R140A) were cloned into pET-47b (Merck Millipore) to produce cleavable His$_6$-tagged fusion proteins. TRIM25-CC (189-379) (UniProt Q14258) was inserted into pET-52b (Merck Millipore) to produce cleavable Strep-Tag II fusion protein. All mutations were introduced by site-directed mutagenesis and verified by DNA sequencing. All proteins were expressed in BL21 (DE3) E.coli cells, induced at 0.6 OD$_{600}$ with 150 μM isopropyl b-D-1-thiogalactopyranoside (IPTG) and incubated at 18 °C for 16 h. Proteins were purified by affinity chromatography, followed by ion-exchange chromatography if necessary (after removal of His, GST or Strep-II tags by HRV 3 C protease) and size-exclusion chromatography (SEC).

Human TRIM25 PRYSPRY (residues 434–630) was cloned from a synthetic gene (GeneArt, Regensburg, Germany) into pFastBacHtb vector for expression in the baculovirus system following standard procedures (Bac-to-Bac manual, Invitrogen). The protein was expressed in HiFive insect cells in Express Five SFM medium (ThermoFisher Scientific, B85502) supplemented with glutamine according to standard protocols. For purification, a cell pellet was resuspended in buffer A (20 mM HEPES pH 7.5, 150 mM NaCl, 20 mM imidazole and 5 mM beta-mercapto-ethanol) and cells were lysed by sonication. Cleared lysate was applied onto a Nickel NTA superflow column (Qiagen). The column was washed with buffer A containing 1 M NaCl and the protein was eluted in buffer A supplemented with 300 mM imidazole. The His-tag was cleaved by incubation with Tobacco Etch Virus (TEV) protease overnight at 4 °C. The protein was dialysed into buffer A and TEV protease and residual his-tagged protein were removed by passage through a Ni-NTA column. Residual aggregates were removed in a final SEC step on a Superdex 200(10/300) column (GE Healthcare) in a buffer containing 20 mM HEPES pH 7.5, 100 mM NaCl, 20 mM imidazole and 5 mM beta-mercapto-ethanol. The protein was concentrated using an AMICON spin concentrator. Human TRIM25 CC-PRYSPRY domain (residues 189–630) was cloned from a synthetic gene (GeneArt, Regensburg, Germany) into pFastBacHtb vector (in between NcoI and HindIII restriction sites) for expression in the baculovirus system (Bac-to-Bac manual, Invitrogen). The protein was expressed in HiFive insect cells in Express Five SFM medium (Invitrogen) supplemented with glutamine according to standard protocols. A cell pellet was resuspended in buffer A (50 mM HEPES pH 7.5, 300 mM NaCl, 20 mM imidazole, 5% glycerol and 10 mM beta-mercapto-ethanol) and cells were lysed by sonication. Cleared lysate was applied onto a Nickel NTA resin (His60 NI Clontech). The column was washed with buffer A containing 1 M NaCl and the protein was eluted in buffer A supplemented with 300 mM imidazole. The His-tag was cleaved by incubation with TEV protease overnight at 4 °C. The protein was dialysed into buffer A (except NaCl was reduced at 120 mM) and TEV protease and residual his-tagged protein were removed by passage through a Ni-NTA column. Cleaved protein was further purified by cation exchange chromatography (HiTrap SP HP, GE Healthcare) a final SEC step on a Superdex 200 Increase (10/300 GL) column (GE Healthcare) in a buffer containing 20 mM HEPES pH 7.5, 100 mM NaCl and 500 μM TCEP. TRIM25 coiled-coil-PRY-SPRY was identified as a dimer (elution volume Ve = 12 ml) confirmed by MALLS (data not shown). 1 mM TCEP was added to the sample and the protein was concentrated using an AMICON spin concentrator.

Full-length TRIM25 was inserted into pIEX-Bac3 vector (Merck Millipore) with an N-terminal His$_{10}$-tag and an HRV 3 C protease cleavage site using ligation independent cloning (LIC). Protein was expressed in SF9 insect cells (ThermoFisher Scientific 1149015), using a protocol described previously[51]. Cells from 2.1 L of culture were resuspended in lysis buffer (100 mM HEPES, pH 7.0, 500 mM NaCl, 5% (v/v) glycerol, 1 mM TCEP, 20 mM Imidazole, 10 mM MgCl$_2$), supplemented with 1 mM PMSF, 2 tablets of EDTA-free protease inhibitor cocktail tablets (Roche), and 25 units/ml Benzonase nuclease (Sigma). Cells were lysed by sonication and debris was removed by centrifugation (55,000 × g for 45 min at 4 °C). The supernatant was incubated with 5 ml of Talon superflow resin (GE Healthcare) for 3 h on a rocker at 4 °C. The resin was washed extensively with wash buffer (100 mM HEPES, pH 7.0, 500 mM NaCl, 10% (v/v) glycerol, 1 mM TCEP, 40 mM Imidazole). Bound protein was eluted in wash buffer containing 300 mM Imidazole in a total volume of 5 ml and incubated with HRV-3C to remove the His$_{10}$-tag. The protein was purified to homogeneity on a HiLoad 16/600 Superdex 200 gel filtration column (GE Healthcare) in size-exclusion buffer (50 mM HEPES, pH 7.0, 100 mM NaCl, 1 mM DTT). Protein-containing fractions were diluted in buffer containing 50 mM HEPES pH 7.0, 50 mM NaCl, 1 mM DTT and applied to a 1 ml SP FF IEX column (GE Healthcare) as a means of protein concentration. Protein was eluted in buffer containing 50 mM HEPES pH 7.0, 300 mM NaCl, 1 mM DTT and flash-frozen in liquid nitrogen and stored at −80 °C.

For stable isotope labelling ($^{15}$N, $^{13}$C, and $^2$H) the human PRYSPRY domain (residues 439-630) was cloned into pETm22 and co-expressed with chaperones

KJE, ClpB and GroELS in *E. coli* BL21(DE3). The protein was purified using Ni-NTA affinity chromatography, prior to cleaving the tag, using 3C protease and removal of the tag by ion-exchange chromatography.

Protein molecular mass was verified by electrospray ionisation mass spectrometry. The fold of wild-type and mutant proteins was analysed by circular dichroism spectroscopy. Protein concentrations were determined by UV absorption at 280 nm using calculated extinction coefficients. Bovine mono-ubiquitin was purchased from Sigma and further purified by SEC.

**Production of the UbcH5c-Ub thioester.** M1C-ubiquitin was labelled with Atto 647 N maleimide (Sigma) as described for Cy5 labelling[50]. UbcH5c~Ub$^{Atto}$ conjugate was synthesised as previously described[52]. Briefly, His-Ube1 (1 μM), UbcH5c (250 μM), Atto 647N-Ub (Ub$^{Atto}$) (500 μM) and ATP (3 mM) (Sigma) were incubated for 60 min at 25 °C. The UbcH5c~Ub$^{Atto}$ thioester-linked conjugate was purified by SEC using a HiLoad 16/60 Sephadex 75 gel filtration column (GE Healthcare) pre-equilibrated in 50 mM HEPES pH 7.5, 150 mM NaCl and flash-frozen in liquid nitrogen.

**In vitro ubiquitination assays.** Ubiquitin discharge assays with pre-charged UbcH5c~Ub$^{Atto}$ were performed as previously described[23]. Briefly, 1 μM UbcH5-c~Ub$^{Atto}$ and 1 μM TRIM25-FL were incubated with 1 or 10 or 100 μM of different NS1 constructs/mutants (NS1-ED, NS1-FL (R38A/K41A/W187A or W187A)) in buffer containing 50 mM HEPES pH 7.5, 150 mM NaCl and 20 mM L-Lysine. Reactions were incubated at 25 °C and quenched with 2× SDS sample buffer at 0, 5, 15 min. Samples were resolved by SDS-PAGE and gels were scanned at 635 nm with a Typhoon FLA 9500 and analysed using ImageQuant (GE Healthcare). Gels were subsequently stained with InstantBlue for total protein content. Ubiquitination assays with UBE2N/UBE2V1 were performed with 0.5 μM His-Ube1, 2.5 μM of each E2, 1 μM TRIM25-FL, 1, 10 or 100 μM of NS1 constructs/mutants (NS1-ED, NS1-FL (R38A/K41A/W187A) or NS1-FL (W187A)), 50 μM ubiquitin and 0.5 μM Ub$^{Atto}$ in buffer containing 50 mM HEPES pH 7.5, 150 mM NaCl, 20 mM MgCl$_2$ and supplemented with 10 mM ATP. Gels were scanned and subsequently stained with InstantBlue. Experiments were performed in triplicate and representative images are shown.

**Cell culture and transfection.** HEK293T cells (supplied by ECACC, authenticated by STR profiling and species ID, tested monthly for mycoplasma contamination) were cultured in Dulbecco's modified Eagle's medium (DMEM; Gibco, Thermo-Fisher Scientific) supplemented with 10% heat-inactivated foetal calf serum (Autogen Bioclear UK, Ltd), 2 mM glutamine and 100 U/ml penicillin/streptomycin (Gibco, ThermoFisher Scientific) at 10% CO$_2$ at 37 °C. 3FLAG-tagged and GST-tagged human RIG-I tandem CARDs (UniProt O95786), NS1 and mutants (strain A/Puerto Rico/8/1934 H1N1) were cloned into pcDNA3.1 and verified by Sanger sequencing. pCMV-HA-TRIM25 was purchased from the MRC-PPU, Dundee (DU 12451). Cells were transfected using Lipofectamine 2000 (Thermo-Fisher) according to manufacturer's instructions.

**RIG-I ubiquitination assays.** HEK293T cells were transfected with 3FLAG-2CARD, TRIM25, and NS1 in a 6-well plate. Twenty-four to 48 h post-transfection, cells were washed once in cold PBS and lysed in plates using an NP-40-based lysis buffer [0.5% NP-40, 150 mM NaCl, 50 mM Tris pH 7.5, 5 mM MgCl$_2$, protease inhibitors (EDTA-free mini tablets, Pierce)] for 45 min at 4 °C. Cell lysates were cleared by centrifugation (14,500 × g, 15 min, 4 °C). Lysates were subjected to FLAG immunoprecipitation using FLAG M2 agarose beads (Sigma Aldrich) for 2 h at 4 °C. Immunoprecipitates were washed three times in lysis buffer and resolved on 4–12% NuPAGE Bis-Tris precast gels (Invitrogen), transferred to PVDF membrane using the iBlot semi-dry transfer system (Invitrogen), blocked in either 1% non-fat dried milk or 2% BSA (Anti-Ubiquitin) in TBS-T (TBS, 0.1% Tween-20), and incubated with the relevant primary antibodies: Anti-FLAG HRP (A8592, Sigma Aldrich, 1:10,000 dilution), Anti-TRIM25 HRP (ab200788, AbCam, 1:5000 dilution), Anti-NS1 HRP (NS1-23–1) (sc130568, Santa-Cruz Biotechnology, 1:500 dilution), Anti-Ubiquitin HRP (PD41) (sc8017, Santa-Cruz Biotechnology, 1:500 dilution). Uncropped scans of western blots are shown in Supplementary Fig. 9.

**Luciferase reporter assay.** $1.25 × 10^5$ HEK293 cells/well were plated in 24-well plates, and transfected the next day with the following constructs: GST-2CARD (25 ng), TRIM25 (10 ng), and NS1 (25/50/100 ng) using empty pcDNA3.1 to adjust the total amount of transfected DNA to 135 ng/well. After 24 h, cells were transfected with 125 ng of p125-Luc, encoding a Firefly luciferase reporter gene under the control of the IFN-β promotor (gift from T. Fujita, Kyoto University, Japan) and 25 ng pRL-TK (Promega) using Lipofectamine 2000. Luciferase activity was measured 18–20 h later using the dual-luciferase assay reporter system (Promega). Firefly luciferase values were divided by Renilla luciferase values to normalise for transfection efficiency. All data are shown as fold increase relative to mock-transfected cells.

**SEC-MALLS.** Analytical SEC-coupled to multi angle laser light scattering (MALLS) profiles were recorded at 16 angles using a DAWN-HELEOS-II laser photometer

(Wyatt Technology) and differential refractometer (Optilab TrEX) equipped with a Peltier temperature-regulated flow cell maintained at 25 °C (Wyatt Technology). Samples of purified proteins at different concentrations (100 μl volume) were applied to a Superdex 200 10/300 GL column (GE Healthcare) equilibrated with 50 mM HEPES pH 7.5, 150 mM NaCl, 0.5 mM TCEP and 3 mM NaN$_3$ at a flowrate of 0.5 ml/min. The data were analysed using ASTRA 6.1.

**Binding studies by biolayer interferometry.** The binding of NS1 constructs to immobilised TRIM25-CC was measured on an Octet RED biolayer interferometer (Pall ForteBio Corp., Menlo Park, CA, USA). His-tagged TRIM25-CC was immobilised on Ni-NTA biosensors (Pall ForteBio Corp., Menlo Park, CA, USA) using a concentration of ~12 μg/ml. The binding of the NS1 constructs (at 1–55 μM) to immobilised TRIM25-CC was measured at 25 °C with a 200 s association step followed by a 200 s dissociation step. The buffer was 50 mM HEPES (pH 7.5), 150 mM NaCl, 1 mM DTT and 0.005% Tween-20. The equilibrium dissociation constant was determined by analysing the variation of the response with construct concentration using non-linear least-squares methods and assuming a simple 1:1 binding model. Experiments were performed at least in triplicate.

**Crystallisation.** NS1-ED (9 mg/ml) was mixed with TRIM25-CC (10 mg/ml) at a 1:1 molar ratio and crystallisation trials set up using an Oryx crystallisation robot. Initial hits were optimised by hanging drop vapour diffusion at 18 °C with a reservoir solution containing 100 mM Tris pH 8.5, 25 % PEG 600 and 200 mM sodium citrate and reaching full size (0.7 mm) after 5 days. Crystals were flash-frozen in the reservoir solution containing 30% PEG 600. NS1-FL (R38A/K41A/W187A) (8 mg/ml) was mixed with TRIM25-CC (10 mg/ml) at a 1:1 molar ratio and crystallisation trials set up using an Oryx crystallisation robot. Clustered plates appeared within 3 days and were used for seeding in order to optimise the size and morphology of the crystals. Single cubic crystals appeared within 3 days in a seeded drop (0.02 μl seed stock + 0.1 μl protein mixture + 0.1 μl reservoir solution) with a reservoir solution containing 100 mM sodium citrate pH 6.5 and 24% PEG 3350. Crystals were cryoprotected in the reservoir solution containing 25% glycerol and flash-frozen.

Human TRIM25 PRYSRPY domain was crystallised by vapour diffusion in sitting drops by mixing 15 mg/ml protein in a 1:1 ratio with reservoir solution containing 0.1 M bis-Tris pH 6.5 and 2 M ammonium sulphate. Crystals grew in 2 days and were cryoprotected by soaking for 30 s in 0.08 M bis-Tris pH 6.5 and 1.6 M ammonium sulphate and 20% glycerol.

Human TRIM25 CC-PRYSPRY was crystallised by vapour diffusion in sitting drop by mixing 10 mg/ml dimer in a 1:1 ratio with reservoir solution. 576 different conditions were initially screened robotically at 20 °C and refined in 24-well hanging drop crystallisation plates by mixing 1 μl protein with 0.5 μl reservoir solution. Large dagger-shaped crystals were obtained after 5 days. The best crystals grew in mother liquor containing 0.2 M sodium formate, 14% PEG 3350 and 0.1 M bis-Tris pH 6.5. 15% glycerol was added to cryo-protect.

**Data collection and structure determination.** Data for both NS1-ED/TRIM25-CC and NS1-FL (R83A/K41/W187A)/TRIM25-CC crystals were collected on beamline IO4 (λ = 0.9795 Å) at the Diamond Light Source (Oxford, UK) and processed using XDS[53] and autoPROC[54], respectively. Both structures were solved by molecular replacement (MR) using 3O9T as a search molecule for NS1-ED, 1AIL for NS1-RBD and 4LTB for TRIM25-CC in Phaser[55]. For the NS1-ED/TRIM25-CC structure, we first identified the MR solution for TRIM25-CC, which was fixed as a partial solution in order to locate NS1-ED. The solution for the three copies of TRIM25-CC and three copies of NS1-ED in the asymmetric unit was unambiguously identified with a translational function Z-score (TFZ) of 19.8. For the NS1-FL (R38A/K41A/W187A)/TRIM25-CC structure, we first identified the partial molecular replacement solution containing two complexes of NS1-ED/TRIM25-CC in the asymmetric unit (TFZ = 16.5). Subsequently, this solution was fixed and two out of four molecules of NS1-RBD were located with a TFZ score of 18.6. Models were iteratively improved by manual building in Coot and refined using REFMAC5 and Phenix[56,57]. All structural figures were prepared in PyMOL. Protein interfaces were calculated using PISA[58]. Coordinates and structure factors were deposited in the Protein Data Bank under accession codes 5NT1 and 5NT2.

For phasing of the human TRIM25 PRYSRPY domain, a crystal grown under native conditions was transferred into the reservoir solution supplemented with 2 mM ethyl-mercuric chloride and soaked for 8 h before cryo-protection and freezing. Diffraction measurements were made at the ESRF on beamlines ID23-2 (native) and ID14-EH4 (Hg derivative) and processed with XDS[53]. The structure was solved by single isomorphous replacement with anomalous scattering. Six mercury sites were found and refined with SHARP[59]. The resultant map was improved by threefold non-crystallographic averaging and an initial model was built by ARP/WARP[60]. Subsequent refinement was done with REFMAC[57]. Data collection and refinement statistics are listed in Supplementary Table 1.

Data of human TRIM25 CC-PRYSPRY crystals were collected on ESRF beamline ID29, which was able to nicely resolve the diffraction spots along the very long c*-axis (1/827 Å$^{-1}$), and processed with XDS[53]. The structure was solved by molecular replacement using PHASER[55] and the known structure of the hTRIM25 coiled-coil (PDB:4LTB)[21] and the hTRIM25 PRYSPRY domain described here.

Refinement was performed with REFMAC[57] and BUSTER. Data collection and refinement statistics are listed in Table 1. An example of the map-sharpened electron density is in Supplementary Figure 4A. There are one and a half coiled-coil dimers in the asymmetric unit with a crystallographic twofold completing the second dimer and the molecules form a three-dimensional network throughout the crystal (Supplementary Figure 4B).

**Small-angle X-ray scattering data collection and analysis**. Small-angle X-ray scattering (SAXS) measurements were carried out at the BM29 beamline at the ESRF, Grenoble[61]. The CC-PRYSPRY sample was measured in NMR buffer (20 mM Na$_2$HPO$_4$ pH 6.5, 150 mM NaCl, 2 mM DTT) at four different concentrations ranging between 1.0 and 20 mg/ml at 300 K. Ten frames with 1 s exposure time per frame were recorded for each sample and buffer using an X-ray wavelength of 0.992 Å. Measurements were performed in flow mode, by pushing the sample through a capillary at constant flowrate to minimise radiation damage. Frames showing radiation damage were removed before data analysis. For SAXS data collection and processing, the dedicated beamline software BsxCuBE was used in an automated fashion. The one-dimensional scattering intensities of samples and buffers were expressed as a function of the modulus of the scattering vector $Q = (4\pi/\lambda)\sin\theta$ with $2\theta$ being the scattering angle and $\lambda$ the X-ray wavelength. Buffer intensities were subtracted from samples using the software PRIMUS[62]. The radii of gyration $R_g$ of all samples were extracted by the Guinier approximation with the same programme and also together with $D_{max}$ were calculated from pairwise distribution functions using GNOM[63]. All statistics are summarised according to Trewhella et al.[64] in Supplementary Table 2.

PRYSPRY-CC models to fit against the experimental data were generated using a restrained MD/SA protocol implemented in CNS[65] as previously described[66]. In short, a template structure with CC and PRYSPRY domain connected with the 73-residue linker was generated. Then the linker (residues 361–433) was kept disordered by random rotation of the $\varphi$ and $\psi$ backbone angles during structure calculations. The resulting structures were subsequently minimised by a 3-step Cartesian dynamics simulated annealing protocol with 40,000 high-temperature (20,000 K) steps followed by two cooling phases of 4000 steps (2000 to 1000 K and 1000 K to 50 K). In a first calculation of 5000 structures, the interface between the CC and PRYSPRY domain was kept rigid but the linker was allowed to move freely. During the second calculation only one PRYSPRY domain was fixed to the CC domain, whereas the other and the linker were allowed to move freely. The third calculation allowed free movement of both PRYSPRY domains including the linker. Each pool of 5000 structures was fitted against SAXS data using CRYSOL[67].

**NMR data collection and analysis**. NMR spectra were acquired at 298 K on a Bruker Avance III NMR spectrometer equipped with a cryogenic triple resonance gradient probe head at a magnetic field strength corresponding to a proton Larmor frequency of 800 MHz. Samples were measured at concentrations ranging between 0.1 and 0.5 mM in 20 mM Na$_2$HPO$_4$, 150 mM NaCl, 0.5 mM TCEP at pH 6.5. All spectra were processed with NMRPipe[68] and analysed using Sparky[69], CCPNMR[70] and CARA (http://cara.nmr.ch). For backbone resonance assignments of the PRYSPRY domain, HNCACB, CBCA(CO)NH and HNCA spectra were recorded. Backbone resonance assignment was obtained to a completion of 95% (175 out of 184 HN-N resonances). Backbone resonance assignments of the Y463S/Y476S double mutant was possible by tracing the peak shifts. Ambiguous peak positions were left out of the analysis. Peak intensities for NMR titrations were calculated in SPARKY by keeping the linewidth constant during fitting and manually corrected for dilution during the titration.

**Data availability**. Atomic coordinates and structure factors have been deposited in the Protein Data Bank under accession codes 5TN1 for the TRIM25-CC/NS1-ED complex, 5NT2 for the TRIM25-CC/NS1-FL complex, 6FLM for the TRIM25 PRYSPRY domain and 6FLN for the TRIM25 CC-PRYSPRY fragment. Backbone assignments of the TRIM25-PRYSPRY domain have been deposited at the Biological Magnetic Resonance Bank (BMRB) under accession code 27381. Other data are available from the corresponding author upon request.

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

## Acknowledgements

We thank the Crick Structural Biology and Proteomics Science Technology Platforms for expert technical support, especially Phil Walker, Andy Purkiss and Evangelos Christodoulou; the Diamond Light Source, Oxford for synchrotron access; Ian Taylor for help with SEC–MALLS experiments; the tutors of the December 2016 DLS/CCP4 Workshop for expert advice; Luigi Martino and all other members of the KR group and Rodolfo Ciuffa, ETH Zürich for helpful discussions, and Steve Smerdon and Ian Taylor for comments on the manuscript. We thank Sandra Augsten for sample preparation and Bernd Simon for structural modelling. We thank the EMBL Grenoble Eukaryotic Expression Facility (EEF) for insect cell culture and HTX Facility for crystallisation. We thank the European Synchrotron Radiation Facility for access to its beamlines for the work on TRIM25 PRYSPRY and CC-PRYSRY domains, Jan Kadlec for help in solving the PRYSPRY structure, Bernd Simon for help in analysing SAXS data and Sandra Anett Augsten for technical help. This work was supported by the Francis Crick Institute which receives its core funding from Cancer Research UK, the UK Medical Research Council and the Wellcome Trust (FC001142 to K.R. and FC001136 to C.R.S.), by a PhD fellowship from the Boehringer Ingelheim Fonds to M.G.K and the Louis-Jeantet Foundation to C.R.S. In addition, this work was supported by ERC grant V-RNA (322586) to S.C. and by the Deutsche Forschungsgemeinschaft (DFG) with an Emmy-Noether Fellowship (HE 7291/1-1) to J.H.

## Author contributions

M.G.K. characterised the TRIM25/NS1 interaction, except experiments in Fig. 1c (S.R. M.) and Figs. 4c and 5a, c (A.G.V. and R.V.S.) and wrote the manuscript. M.L. expressed and crystallised the TRIM25 CC-PRYSPRY domain. K.H. and J.H. performed and analysed SAXS and NMR experiments. E.K. crystallised and solved the structure of the TRIM25 PRYSPRY domain. C.R.S. analysed data. S.C. and K.R. analysed data, supervised the project and wrote the manuscript.

## Additional information

**Competing interests:** The authors declare no competing interests.

