## [Peer Review File · Nature Communications]

Reviewers' comments:

Reviewer #1 (Remarks to the Author):

RNA virus infections are predominantly detected by RIG-I-like receptors (RLR), mainly RIG-I and MDA5. Activation of RLRs leads to the initiation of downstream signalling cascades, culminating in the induction of type I interferon and other cytokines. Activation of RLRs is regulated to prevent aberrant activation of innate immunity. For example, RIG-I is modified by K63-linked polyubiquitylation induced by the E3 ligase TRIM25, which leads to RIG-I activation. The non-structural protein NS1 of influenza A viruses was previously shown to bind TRIM25, preventing RIG-I ubiquitination and thereby downstream signalling. The binding sites of TRIM25 and NS1 have been mapped (coiled-coil region in TRIM25 and E96/E97-surrounding region in NS1), and it was previously suggested that NS1 binding results in defective oligomerization of TRIM25.

Koliopoulos and Rittinger now solved the crystal structure of the TRIM25 coiled coil domain in complex with the effector of NS1 as well as TRIM25 coiled coil domains in complex with a triple mutant of NS1 full length, which largely confirms previous work and also provides additional information into this interaction. Furthermore, the topic of this study will be of high interest to the fields of innate immunity and influenza. However, the major weakness of this study is that the authors used for most of their analyses a mutant version of NS1 (R38A, K41A, W187A) containing the R38A and K41A mutations, which are well known to abrogate NS1's ability to block IFN induction, TRIM25 binding and RIG-I inhibition (e.g. Donelan NR et al., J Virol 2003, Gack MU et al., CHM 2009, Talon J et al., J Virol 2000). Using *in vitro* assays, the authors provide results that contradict work by several research groups, which characterized in detail this NS1 mutant using experiments in cells, and even recombinant mutant influenza A viruses. Therefore, there is the overall concern that the *in vitro* functional analyses performed in this study, in which the authors found that this mutant still binds to TRIM25 and blocks RIG-I, is an artifact of the experimental system. Furthermore, wild-type NS1 was not included in any of the functional assays. Although we acknowledge that it may be difficult to purify wild-type NS1, including WT NS1 in the functional assays (TRIM25 binding, ubiquitylation, and dimerization) is required to draw conclusions. Moreover, functional data in cells (e.g. ubiquitylation assays for RIG-I, induction of type I interferon and its inhibition by NS1) is required to complement the *in vitro* assays.

Additional major comments:

Fig. 1C: Can the authors comment on why the double mutant (R38A and K41A) shows increased binding, although less protein was used in the input? WT NS1 should be included as a positive control in this assay.

Fig. 3C: To convincingly show that the identified protein-protein contacts in the crystal structure are true interaction sites, the authors need to mutate not only amino acids in NS1, but also test mutants of the TRIM25-CC.

Fig. 3C: It would be important to verify that loss of binding correlates with a loss of function in cell culture-based systems.

Fig. 5: WT NS1 protein needs to be added to these assays as control. Moreover, since non-binding mutants were identified in Fig. 3, they should also be included as negative controls in the functional assays.

Fig. 6A: RIG-I activation requires K63-linked polyubiquitylation in infected cells. The authors here investigate RIG-I mono-ubiquitination in vitro, which is unlikely to reflect the physiological situation. The authors should test RIG-I K63-linked ubiquitylation and its inhibition by NS1 in cell culture-based assays. Additionally, WT NS1 should be included in these assays.

Fig. 6: The authors used an oligomerization-defective mutant of NS1 to draw the conclusion that oligomerization of TRIM25 by NS1 binding is not impaired. Changes in TRIM25 oligomerization by NS1 needs to be addressed by using wildtype NS1. The authors should also test the effect of NS1 WT and mutants on TRIM25 oligomerization using gel purification or native PAGE.

Fig. 7: The authors propose a very interesting model involving a putative movement of the SPRY domain; however, no data are provided to confirm this model, and as such, the figure of the model should be removed and instead discussed in the discussion section. Alternatively, the authors need to provide biochemical data to support this model. For example, they could test whether SPRY/coiled-coiled interactions are not observed anymore upon NS1 association. In these assays WT NS1 needs to be included.

Minor comments:

Fig. 1 A: Figure labeling should clarify which proteins are displayed.

Fig. 3A and B: The color-coding and illustration of the binding interface could be visually improved by highlighting the important interactions as described in the main text.

Fig. 4: The labeling is misleading as it should be pointed out that the NS1-FL is in fact a triple mutant of NS1.

Fig. 5A: What are the background bands seen in the assay for the triple and single mutants of NS1-FL?

The legends of the Suppl. Figures do contain minimal information about the experiments. The authors should add more information to the legends.

Reviewer #2 (Remarks to the Author):

In this study Koliopoulos and Rittinger describe crystal structures of TRIM25 in complex with

NS1 from influenza A. More specifically, the coiled-coiled of TRIM25 (TRIM25-CC) was crystallized in complex with either full-length NS1 or with the effector domain (ED) of NS1 only. Because wild-type NS1 forms aggregates in solution, mutant NS1 variants were used, with either the R38A/K41A double mutation (previously established by Gack et al), with a novel W187A mutation, or with all three mutations. The authors show that all three NS1 variants bind to TRIM25, which was unexpected in light of a previous proposal by Gack et al that the R38A/K41A double mutation interfered with TRIM25-CC binding. The crystal structures show that NS1 binds to TRIM25 via a partially hydrophobic interface that includes Leu95 in NS1-ED without perturbing CC-mediated dimerization of TRIM25. This was unexpected as previous reports had suggested that NS1 might inhibit TRIM25-dependent ubiquitination of RIG-I by interfering with TRIM25 dimerization, which is required for ubiquitination activity. Moreover, the authors show that binding of NS1 does not affect the E3 ligase activity of TRIM25 using a UbcH5-ubiquitin discharge assay. Generation of unanchored ubiquitin chains by Ubc13/UEV1 with TRIM25 is also unaffected by NS1. However, the authors found that full-length NS1 did reduce mono-ubiquitination of the tandem CARDs of RIG-I by Ubc13 together with TRIM25 by 55% (the NS1-ED domain alone had no effect). The authors conclude that binding of NS1 to TRIM25 prevents ubiquitination of RIG-I through steric effects (eg. by preventing the RIG-I CARDs getting close enough to the TRIM25-RING domain and E2 ligase to be ubiquitinated), without directly affecting the catalytic properties of the E2/E3 ubiquitin ligase complex.

This manuscript is clearly and thoughtfully written, and provides new insight on the proviral activity of NS1 in antagonizing K63-linked ubiquitination of RIG-I during antiviral signaling. The data presented rule out the previously proposed mechanisms of inhibition of TRIM25 dimerization, and inhibition of E2/E3 ligase activity of the TRIM25/E2 complex by NS1. The main weakness of the manuscript, is that the actual function of NS1 remains unclear. The extent to which the moderate decrease in mono-ubiquitination of RIG-I CARDs by TRIM25 in the presence of NS1 accounts for the physiological function of NS1 is questionable, particularly given that NS1 and TRIM25 itself are also mono-ubiquitinated in the Ubc13-dependent assay, which moreover utilizes a non-physiological E2 ligase. The resulting conclusion that "NS1-binding to TRIM25 inhibits RIG-I ubiquitination by steric effects" is not very informative or satisfying. It is tempting for this reviewer to conclude that the natural tendency of wild-type NS1 to aggregate may hold the key to the proviral activity of NS1, and unfortunately the current study does not address this question directly as only the mutant NS1 variants with increased solubility could be studied with the current set of biochemical and biophysical approaches.

Major concerns

1. The hypothesis that NS1 binding to TRIM25 restricts K63-ubiquitination of RIG-I, and hence antiviral signaling, through steric effects is vague and needs confirmation and further development. To fully answer this question will require work that will extend beyond a reasonable scope for the current study, but some validation for the steric hindrance hypothesis would increase the impact of this study. For example, steric inhibition should be achievable with other proteins of similar size to NS1- can TRIM25 ubiquitination of RIG-I CARD be inhibited by binding of a protein other than NS1 to a similar site on TRIM25 (eg. another TRIM25 binding protein or antibody)? Or, is the effect still observed when the native

heterodimeric E2 is used instead of Ubc13? Does increasing the bulk of NS1 with an exogenous fusion protein enhance its inhibitory activity by increasing the steric effect?

2. The pull-down experiments shown in Fig. 1 are satisfactory to give a yes/no answer regarding binding of NS1 to TRIM25, but the authors should ideally measure binding curves and Kds for TRIM25-NS1 binding. Since each construct was purified in large quantities for SEC-MALLS experiments, one would assume that it would be straightforward for these authors to generate binding curves using a standard binding assay (eg. SPR, ITC, BLI or DSF).

Minor points

1. The last Results subheading, "Mechanism of NS1-mediated RIG-I inhibition" is somewhat misleading as the mechanism has not been fully elucidated. The subheading should be changed to something more descriptive such as "NS1 binding to TRIM25 restricts K63 mono-ubiquitination of RIG-I CARDS".

2. The repeated statement that dimerization of the TRIM25 RING domains occurs in an intramolecular fashion (p. 4, p. 11, p. 15) is confusing. How can a protein form dimers in an intramolecular fashion? It appears from Fig. 7 that the authors mean that the RING-Box domains are thought to fold back on the CC domain of the same molecule during dimerization, but this should be clarified throughout the text.

3. P. 7 (top). How do the binding affinities of FL-NS1 and NS1-ED for TRIM25 compare? Does NS1-RBD have any residual binding affinity for TRIM25 in an equilibrium binding assay (eg. ITC)?

4. P. 5, 10, 12, 13. Remove the commas after "both".

5. P. 10. Change "...important for its function" to "...important for the proviral function of NS1", or similar, to clarify.

6. In the discussion, the likely role of wild-type NS1 aggregation in its inhibition of TRIM25-dependent ubiquitination of RIG-I should be further emphasized. Does wild-type NS1 inhibit unanchored ubiquitin chain production?

7. P. 17. Is the TRIM25 construct used a human sequence? Please specify.

8. Fig. 2B. This panel is not very effective at showing the two interfaces and how they differ from each other. Please consider changing the view or representation. Panels A and B should also both be enlarged with less blank space around them.

9. Fig 3A, B. The panels should be enlarged with less blank space around them. Shadowing should be removed (A). Depth cueing should be added to A. The contrast of the labels should be enhanced in B.

10. Figures 2 and 3 could be consolidated into a single figure.

11. Fig. S2C. Is the NS1 dimer shown in an arbitrary position? This panel is confusing and it is not obvious what point it is trying to make.

12. Fig. S5. The two views should be adjusted so they are on exactly the same scale.

We thank the reviewers for their constructive comments. To address the concerns raised we have significantly extended our study and now provide additional validation of our structural work through quantitative protein-protein interaction data and ubiquitination and luciferase-based IFN assays in cells. Importantly, we present the novel structure of a TRIM25 CC-PRYSPRY fragment. Comparison with our TRIM25 CC/NS1 complex structures shows that NS1 and the PRYSPRY domain have overlapping sites on the CC and cannot bind simultaneously. Moreover, we show that direct contacts between the TRIM25 CC and PRYSPRY domains are crucial for RIG-I ubiquitination. These additional results strongly support our model that NS1 inhibits RIG-I ubiquitination through steric effects. In addition, the direct binding of dimeric NS1 to TRIM25 coupled with the self-association of NS1 would lead to sequestration of TRIM25 in higher order complexes and this is likely a key feature of the proviral mechanism of NS1, as proposed by reviewer #2.

Below is a detailed point-by point response to the concerns raised.

Reviewer #1 (Remarks to the Author):

RNA virus infections are predominantly detected by RIG-I-like receptors (RLR), mainly RIG-I and MDA5. Activation of RLRs leads to the initiation of downstream signalling cascades, culminating in the induction of type I interferon and other cytokines. Activation of RLRs is regulated to prevent aberrant activation of innate immunity. For example, RIG-I is modified by K63-linked polyubiquitylation induced by the E3 ligase TRIM25, which leads to RIG-I activation. The non-structural protein NS1 of influenza A viruses was previously shown to bind TRIM25, preventing RIG-I ubiquitination and thereby downstream signalling. The binding sites of TRIM25 and NS1 have been mapped (coiled-coil region in TRIM25 and E96/E97-surrounding region in NS1), and it was previously suggested that NS1 binding results in defective oligomerization of TRIM25.

Koliopoulos and Rittinger now solved the crystal structure of the TRIM25 coiled coil domain in complex with the effector of NS1 as well as TRIM25 coiled coil domains in complex with a triple mutant of NS1 full length, which largely confirms previous work and also provides additional information into this interaction. Furthermore, the topic of this study will be of high interest to the fields of innate immunity and influenza. However, the major weakness of this study is that the authors used for most of their analyses a mutant version of NS1 (R38A, K41A, W187A) containing the R38A and K41A mutations, which are well known to abrogate NS1's ability to block IFN induction, TRIM25 binding and RIG-I inhibition (e.g. Donelan NR et al., J Virol 2003, Gack MU et al., CHM 2009, Talon J et al., J Virol 2000). Using in vitro assays, the authors provide results that contradict work by several research groups, which characterized in detail this NS1 mutant using experiments in cells, and even recombinant mutant influenza A viruses. Therefore, there is the overall concern that the in vitro functional analyses performed in this study, in which the authors found that this mutant still binds to TRIM25 and blocks RIG-I, is an artifact of the experimental system. Furthermore, wild-type NS1 was not included in any of the functional assays. Although we acknowledge that it may be difficult to purify wild-type NS1, including WT NS1 in the functional assays (TRIM25 binding, ubiquitylation, and dimerization) is required to draw conclusions. Moreover, functional data in cells (e.g. ubiquitylation assays for RIG-I, induction of type I interferon and its inhibition by NS1) is required to complement the in vitro assays.

We appreciate that our data may seem to contradict previous studies that showed that mutations R38A and K41A in NS1 abrogate its ability to block IFN production. However, we believe that our data instead extend previous studies and provide important new insight into the mechanisms by which NS1 interferes with host activities. The study by Donelan 2003,

showed that the R38A/K41A mutation in NS1 interfered with binding to dsRNA and that this activity is required for its IFN antagonist properties – something we do not contradict. In contrast, our structures show that the region around R38A and K41A is not involved in binding to TRIM25 and hence that the dsRNA- and TRIM25-binding regions do not overlap. The previous suggestion that binding of NS1 to TRIM25 interferes with the dimerization of the coiled coil (CC) domain of TRIM25 was made before structures of TRIM protein CC regions became available. Structures of the CC domains of TRIM25, TRIM20, TRIM69 and TRIM5 α now show that they all have highly similar CC arrangements that are antiparallel, span around 170 Å in length and bury over 5000 Å² of surface area. To disrupt such an extensive interface would be energetically very costly and would be extremely difficult to achieve through a protein-protein interaction. Instead, our data provide strong structural and functional support for an alternative mechanism by which NS1 can interfere with the activity of TRIM25: by preventing the substrate-binding PRYSPRY domain adopting its correct position on the CC domain, which we show is required for efficient substrate ubiquitination.

Crucially, our structures also reveal that the surface of NS1 used to contact TRIM25 overlaps to a significant degree with the surface NS1 uses to bind another host protein, p85 β . This provides important new insight into the mechanism of NS1 and highlights that great care needs to be taken when assessing the effect of point mutations in recombinant virus assays as a given mutation might affect more than one host function.

Additional major comments:

Fig. 1C: Can the authors comment on why the double mutant (R38A and K41A) shows increased binding, although less protein was used in the input? WT NS1 should be included as a positive control in this assay.

As requested by reviewer 2 we have now characterised the TRIM25/NS1 interaction using biolayer interferometry (BLI) to provide a quantitative measure of complex formation and have replaced the original pull-down figures. The BLI experiments do not show significant differences between the mutants tested. Importantly, these quantitative binding studies show that the isolated effector domain can still bind TRIM25, whereas we could not detect an interaction between TRIM25-CC and the isolated RBD up to 300 μ M protein.

Fig. 3C: To convincingly show that the identified protein-protein contacts in the crystal structure are true interaction sites, the authors need to mutate not only amino acids in NS1, but also test mutants of the TRIM25-CC.

Our X-ray structures and binding studies show that interactions between Leu95 from NS1 and a hydrophobic pocket of the CC are important for complex formation. The residues involved in forming the hydrophobic pocket of TRIM25 are likely also involved in stabilizing the coiled coil and hence mutation of such residues would likely destabilize the CC and have secondary effects, making it difficult to interpret binding studies.

Fig. 3C: It would be important to verify that loss of binding correlates with a loss of function in cell culture-based systems.

As requested, we have carried out RIG-I ubiquitination experiments and RIG-I-2CARD-induced expression of the IFN- β promoter-regulated luciferase reporter in cell culture-based assays. These experiments are shown in Figure 5.

Fig. 5: WT NS1 protein needs to be added to these assays as control. Moreover, since non-

binding mutants were identified in Fig. 3, they should also be included as negative controls in the functional assays.

WT NS1 is now included in all cell-based assays. We have tested the L95A/S99A mutant that did not bind TRIM25 in the context of the isolated effector domain, as well as the triple Y89A/L95A/S99A mutant in full-length NS1 in the functional assays. As described in the manuscript these mutant NS1 proteins are still able to interfere with TRIM25-mediated ubiquitination of RIG-I, which we believe is a strong indication that self-association of NS1 and NS1-mediated formation of higher order TRIM25-NS1 assemblies is a key factor contributing to the physiological role of NS1 and which may overcome an otherwise low affinity interaction by avidity effects.

Fig. 6A: RIG-I activation requires K63-linked polyubiquitylation in infected cells. The authors here investigate RIG-I mono-ubiquitination in vitro, which is unlikely to reflect the physiological situation. The authors should test RIG-I K63-linked ubiquitylation and its inhibition by NS1 in cell culture-based assays. Additionally, WT NS1 should be included in these assays.

As requested, we have carried out these assays, which are now presented in Figure 5.

Fig. 6: The authors used an oligomerization-defective mutant of NS1 to draw the conclusion that oligomerization of TRIM25 by NS1 binding is not impaired. Changes in TRIM25 oligomerization by NS1 needs to be addressed by using wildtype NS1. The authors should also test the effect of NS1 WT and mutants on TRIM25 oligomerization using gel purification or native PAGE.

The model proposed by Gack and colleagues in 2009 that binding of NS1 would disrupt the coiled coil-mediated dimerization of TRIM25 was put forward before the first structures of TRIM protein CC domains became available. It is now well established that the CC region of TRIM proteins forms an extensive ~ 170 Å long interface that buries over 5000 Å² of surface area (see for example Sanchez et al., 2014, PNAS, 2494-2499). It is difficult to imagine how it would energetically be possible to disrupt such an extensive interface through a simple protein-protein interaction.

Importantly, our two structures of TRIM25-CC/NS1 complexes show that the CC interface is not disrupted by NS1 and instead it is the position of the linker connecting helices $\alpha 2$ and $\alpha 3$ that is changed, which in turn interferes with the positioning of the PRYSPRY domain and RIG-I ubiquitination.

Fig. 7: The authors propose a very interesting model involving a putative movement of the SPRY domain; however, no data are provided to confirm this model, and as such, the figure of the model should be removed and instead discussed in the discussion section. Alternatively, the authors need to provide biochemical data to support this model. For example, they could test whether SPRY/coiled-coiled interactions are not observed anymore upon NS1 association. In these assays WT NS1 needs to be included.

In the new manuscript we present the structure of the TRIM25 CC-PRYSPRY fragment, which fully supports the model suggested and shows that binding of the PRYSPRY and NS1 are mutually exclusive. In addition we have made structure-based mutations that disrupt the CC/PRYSPRY interface and tested these in cell-based ubiquitination assays. These show that the observed CC/PRYSPRY interface is required for RIG-I ubiquitination.

Minor comments:

Fig. 1 A: Figure labeling should clarify which proteins are displayed.

Done as requested.

Fig. 3A and B: The color-coding and illustration of the binding interface could be visually improved by highlighting the important interactions as described in the main text.

This has been done. Please note that Figures 2 and 3 are now combined into a single Figure (Fig.2)

Fig. 4: The labeling is misleading as it should be pointed out that the NS1-FL is in fact a triple mutant of NS1.

Former Figure 4 is now Figure 3 and we now say explicitly in the text and figure legend that the NS1 protein used is the triple mutant.

Fig. 5A: What are the background bands seen in the assay for the triple and single mutants of NS1-FL?

The background bands are auto-ubiquitinated TRIM25 and ubiquitinated NS1. This is now explicitly labelled in the figure and highlighted in the figure legend. Note please, that this figure has been moved to Supplementary material and we now only show the K63 polyubiquitin chain formation assays in the main manuscript.

The legends of the Suppl. Figures do contain minimal information about the experiments. The authors should add more information to the legends.

We have extended the text and have added additional Figures.

Reviewer #2 (Remarks to the Author):

In this study Koliopoulos and Rittinger describe crystal structures of TRIM25 in complex with NS1 from influenza A. More specifically, the coiled-coiled of TRIM25 (TRIM25-CC) was crystallized in complex with either full-length NS1 or with the effector domain (ED) of NS1 only. Because wild-type NS1 forms aggregates in solution, mutant NS1 variants were used, with either the R38A/K41A double mutation (previously established by Gack et al), with a novel W187A mutation, or with all three mutations. The authors show that all three NS1 variants bind to TRIM25, which was unexpected in light of a previous proposal by Gack et al that the R38A/K41A double mutation interfered with TRIM25-CC binding. The crystal structures show that NS1 binds to TRIM25 via a partially hydrophobic interface that includes Leu95 in NS1-ED without perturbing CC-mediated dimerization of TRIM25. This was unexpected as previous reports had suggested that NS1 might inhibit TRIM25-dependent ubiquitination of RIG-I by interfering with TRIM25 dimerization, which is required for ubiquitination activity. Moreover, the authors show that binding of NS1 does not affect the E3 ligase activity of TRIM25 using a UbcH5-ubiquitin discharge assay. Generation of unanchored ubiquitin chains by Ubc13/UEV1 with TRIM25 is also unaffected by NS1. However, the authors found that full-length NS1 did reduce mono-ubiquitination of the tandem CARDS of RIG-I by Ubc13

together with TRIM25 by 55% (the NS1-ED domain alone had no effect). The authors conclude that binding of NS1 to TRIM25 prevents ubiquitination of RIG-I through steric effects (eg. by preventing the RIG-I CARDS getting close enough to the TRIM25-RING domain and E2 ligase to be ubiquitinated), without directly affecting the catalytic properties of the E2/E3 ubiquitin ligase complex.

This manuscript is clearly and thoughtfully written, and provides new insight on the proviral activity of NS1 in antagonizing K63-linked ubiquitination of RIG-I during antiviral signaling. The data presented rule out the previously proposed mechanisms of inhibition of TRIM25 dimerization, and inhibition of E2/E3 ligase activity of the TRIM25/E2 complex by NS1. The main weakness of the manuscript, is that the actual function of NS1 remains unclear. The extent to which the moderate decrease in mono-ubiquitination of RIG-I CARDS by TRIM25 in the presence of NS1 accounts for the physiological function of NS1 is questionable, particularly given that NS1 and TRIM25 itself are also mono-ubiquitinated in the Ubc13-dependent assay, which moreover utilizes a non-physiological E2 ligase. The resulting conclusion that “NS1-binding to TRIM25 inhibits RIG-I ubiquitination by steric effects” is not very informative or satisfying. It is tempting for this reviewer to conclude that the natural tendency of wild-type NS1 to aggregate may hold the key to the proviral activity of NS1, and unfortunately the current study does not address this question directly as only the mutant NS1 variants with increased solubility could be studied with the current set of biochemical and biophysical approaches.

We thank this reviewer for his/her positive comments. We agree that in our previous submission our steric hindrance model was largely speculative as we did not have supporting structural data. We now present the structure of a TRIM25 CC-PRYSRY fragment, which fully supports the steric hindrance model. Nevertheless, we fully agree with this reviewer that the natural tendency of wild-type NS1 to aggregate is likely to be a crucial part of the activity of NS1. We explicitly discuss this in the manuscript.

Major concerns

1. *The hypothesis that NS1 binding to TRIM25 restricts K63-ubiquitination of RIG-I, and hence antiviral signaling, through steric effects is vague and needs confirmation and further development. To fully answer this question will require work that will extend beyond a reasonable scope for the current study, but some validation for the steric hindrance hypothesis would increase the impact of this study. For example, steric inhibition should be achievable with other proteins of similar size to NS1- can TRIM25 ubiquitination of RIG-I CARD be inhibited by binding of a protein other than NS1 to a similar site on TRIM25 (eg. another TRIM25 binding protein or antibody)? Or, is the effect still observed when the native heterodimeric E2 is used instead of Ubc13? Does increasing the bulk of NS1 with an exogenous fusion protein enhance its inhibitory activity by increasing the steric effect?*

As described above the new TRIM25 CC-PRYSRY fragment structure fully supports our model and highlights intramolecular contacts between the CC region and the PRYSRY domain that are important for RIG-I ubiquitination, as shown in the ubiquitination assay in Figure 4c. These intramolecular contacts are disrupted upon NS1 interaction with the CC which displaces the linker connecting helices $\alpha 2$ and $\alpha 3$ towards the PRYSRY binding site, as shown in Figure 4d.

2. *The pull-down experiments shown in Fig. 1 are satisfactory to give a yes/no answer regarding binding of NS1 to TRIM25, but the authors should ideally measure binding curves and Kds for TRIM25-NS1 binding. Since each construct was purified in large quantities for*

SEC-MALLS experiments, one would assume that it would be straightforward for these authors to generate binding curves using a standard binding assay (eg. SPR, ITC, BLI or DSF).

These quantitative binding experiments have been done as requested (using BLI) and these new experiments now replace the pull-downs shown in the original submission in Figure 1.

Minor points

1. The last Results subheading, "Mechanism of NS1-mediated RIG-I inhibition" is somewhat misleading as the mechanism has not been fully elucidated. The subheading should be changed to something more descriptive such as "NS1 binding to TRIM25 restricts K63 mono-ubiquitination of RIG-I CARDS".

Please see our reply to point 1.

2. The repeated statement that dimerization of the TRIM25 RING domains occurs in an intramolecular fashion (p. 4, p. 11, p. 15) is confusing. How can a protein form dimers in an intramolecular fashion? It appears from Fig. 7 that the authors mean that the RING-Box domains are thought to fold back on the CC domain of the same molecule during dimerization, but this should be clarified throughout the text.

This is exactly what we mean and we have tried to clarify in the revised manuscript if we are talking about dimerization mediated by the anti-parallel CC or of dimerization between the RING domains, which is strictly required for E3 ligase activity and may occur in an intramolecular fashion (our model) or an intermolecular fashion through the formation of higher order TRIM25 assemblies.

3. P. 7 (top). How do the binding affinities of FL-NS1 and NS1-ED for TRIM25 compare? Does NS1-RBD have any residual binding affinity for TRIM25 in an equilibrium binding assay (eg. ITC)?

As shown in the BLI experiments of Figure 1c, we cannot detect any residual binding of the free RBD to immobilized TRIM25-CC up to 300 μ M protein. The affinity of the isolated ED is in a similar range to those of the FL NS1 mutants tested. However, this likely does not represent the situation with wild-type FL NS1, which we would expect to be significantly higher due to avidity effects of higher order oligomerization. All the full-length NS1 proteins tested are mutants in which higher order self-association/aggregation had to be suppressed to produce a "well behaved" protein that could be analysed in BLI binding experiments. Furthermore, the spacing of TRIM25-CC on the biosensor tip is probably too far apart to allow the NS1-mediated crosslinking of TRIM25 that we observed in the full-length NS1/TRIM25-CC crystal structure. We believe that this explains the similarity in dissociation constants measured.

4. P. 5, 10, 12, 13. Remove the commas after "both".

Done

5. P. 10. Change "...important for its function" to "...important for the proviral function of NS1", or similar, to clarify.

Done

6. In the discussion, the likely role of wild-type NS1 aggregation in its inhibition of TRIM25-dependent ubiquitination of RIG-I should be further emphasized. Does wild-type NS1 inhibit unanchored ubiquitin chain production?

We have emphasized the potential role of wild-type NS1 aggregation on TRIM25-dependent ubiquitination of RIG-I in the Discussion.

We agree that it would be ideal to investigate if WT NS1 inhibits unanchored ubiquitin chain production. Unfortunately, we cannot think of an experimental set-up to test this possibility. In a cell-based experiment it would be impossible to exclude that another E3 ligase such as Riplet is responsible for the formation of unanchored K63-linked chains. On the other hand we cannot work with WT NS1 in a recombinant assay, which is in our opinion to only set-up that would unambiguously show that any chains detected are synthesized by TRIM25.

7. P. 17. Is the TRIM25 construct used a human sequence? Please specify.

Done. It is human.

8. Fig. 2B. This panel is not very effective at showing the two interfaces and how they differ from each other. Please consider changing the view or representation. Panels A and B should also both be enlarged with less blank space around them.

The panels have been enlarged but we kept the original orientation to show the two-fold symmetry required to produce the TRIM25 CC dimer.

9. Fig 3A, B. The panels should be enlarged with less blank space around them. Shadowing should be removed (A). Depth cueing should be added to A. The contrast of the labels should be enhanced in B.

We have changed this Figure and hope it is now clearer.

10. Figures 2 and 3 could be consolidated into a single figure.

Done

11. Fig. S2C. Is the NS1 dimer shown in an arbitrary position? This panel is confusing and it is not obvious what point it is trying to make.

We agree that this Figure wasn't helpful and we have therefore removed it.

12. Fig. S5. The two views should be adjusted so they are on exactly the same scale.

We have rescaled this Figure as requested, note please this Figure is now Fig. S7.

REVIEWERS' COMMENTS:

Reviewer #1 (Remarks to the Author):

Although the authors extended their study during the revision, the major weakness of the study, the utilization of mutant NS1 variants defective in self-aggregation and interferon antagonism for the crystal structure analysis and the majority of other experiments, still remains. It remains to be determined whether wild-type NS1 affects TRIM25 activity in the same way.

Reviewer #2 (Remarks to the Author):

The addition to this study of a the new TRIM25 CC-PRYSRY fragment structure showing that the binding site on TRIM25 overlaps with the NS1 binding site on TRIM25 addresses my major concern about the notion of NS1-mediated steric hindrance being too vague and insufficiently developed in the first version of the manuscript. The new structure does indeed supports the model that NS1 binding ubiquitination by interfering with ubiquitin ligase recognition.

The new quantitative binding data for TRIM25-NS1 binding address my second major concern.

The more general concern shared with Reviewer 1 that the natural tendency of wild-type NS1 to aggregate, which is not directly addressed in this study, may hold the key to the proviral activity of NS1, remains. Nevertheless, the increased emphasis on this topic in the Discussion is helpful.